# What can Large Language Models do in chemistry? A comprehensive benchmark on eight tasks

**Taicheng Guo**,* **Kehan Guo**,* **Bozhao Nan, Zhenwen Liang, Zhichun Guo,**
**Nitesh V. Chawla, Olaf Wiest, Xiangliang Zhang**[†]
University of Notre Dame
{tguo2, kguo2, bnan, zliang6, zguo5, nchawla, owiest, xzhang33}@nd.edu

## Abstract

Large Language Models (LLMs) with strong abilities in natural language process-ing tasks have emerged and have been applied in various kinds of areas such as science, finance and software engineering. However, the capability of LLMs to advance the field of chemistry remains unclear. In this paper, rather than pursuing state-of-the-art performance, we aim to evaluate capabilities of LLMs in a wide range of tasks across the chemistry domain. We identify three key chemistry-related capabilities including understanding, reasoning and explaining to explore in LLMs and establish a benchmark containing eight chemistry tasks. Our anal-ysis draws on widely recognized datasets facilitating a broad exploration of the capacities of LLMs within the context of practical chemistry. Five LLMs (GPT-4, GPT-3.5, Davinci-003, Llama and Galactica) are evaluated for each chemistry task in zero-shot and few-shot in-context learning settings with carefully selected demonstration examples and specially crafted prompts. Our investigation found that GPT-4 outperformed other models and LLMs exhibit different competitive levels in eight chemistry tasks. In addition to the key findings from the comprehen-sive benchmark analysis, our work provides insights into the limitation of current LLMs and the impact of in-context learning settings on LLMs' performance across various chemistry tasks. The code and datasets used in this study are available at https://github.com/ChemFoundationModels/ChemLLMBench.

## 1 Introduction

Large language models (LLMs) have recently demonstrated impressive reasoning abilities across a wide array of tasks. These tasks are not limited to natural language processing, but also extend to various language-related applications within scientific domains [56, 30, 24, 10]. Much of the research on the capacity of LLMs in science has been focused on tasks such as answering medical [30] and scientific questions [24, 25]. However, the exploration of their application to practical tasks in the field of chemistry remains underinvestigated. Although some studies [6, 27, 63, 48] have been conducted, they tend to focus on specific case studies rather than a comprehensive or systematic evaluation. The exploration of LLMs' capabilities within the field of chemistry has the potential to revolutionize this domain and expedite research and development activities [62]. Thus, the question, ***"What can LLMs do in chemistry?"*** is a compelling topic of inquiry for both AI researchers and chemists. Nevertheless, there exist two challenges that hinder the answer to the topic and the further development of LLMs in chemistry:

---

*Both authors contribute equally to the work, under the support of NSF Center for Computer Assisted Synthesis (C-CAS). https://ccas.nd.edu.

[†]Corresponding author.

37th Conference on Neural Information Processing Systems (NeurIPS 2023) Track on Datasets and Benchmarks.

- Determining the potential capabilities of LLMs in chemistry requires a systematic analysis of both LLMs and the specific requirements of chemistry tasks. There are different kinds of tasks in chemistry, some of which can be formulated to tasks solved by LLMs while others may not. It is necessary to consider the specific knowledge and reasoning required for each task and assess whether LLMs can effectively acquire and utilize that knowledge.

- Conducting reliable and wide-ranging evaluation requires diverse experimental settings and limitations, that is, careful consideration and standardization of evaluation procedures, dataset curation, prompt design, and in-context learning strategies. Additionally, the API call time consumption and the randomness of LLMs limit the size of the testing.

To address this knowledge gap, we (a group of AI researchers and chemists) have developed a comprehensive benchmark to provide a preliminary investigation into the abilities of LLMs across a diverse range of practical chemistry tasks. Our aim is to gain insights that will be beneficial to both AI researchers and chemists to advance the application of LLMs in chemistry. For AI researchers, we provide insights into the strengths, weaknesses, and limitations of LLMs in chemistry-related tasks, which can inform the further development and refinement of different AI techniques for more effective applications within the field. For chemists, our study provides a better understanding of the tasks in which they can rely on current LLMs. Utilizing our more extensive experimental setup, a broader range of chemistry tasks can be explored to further evaluate the capabilities of LLMs.

Our investigation focuses on 8 practical chemistry tasks, covering a diverse spectrum of the chemistry domain. These include: 1) name prediction, 2) property prediction, 3) yield prediction, 4) reaction prediction, 5) retrosynthesis (prediction of reactants from products), 6) text-based molecule design, 7) molecule captioning, and 8) reagents selection. Our analysis draws on widely available datasets including BBBP, Tox21 [65], PubChem [32], USPTO [29, 53, 39], and ChEBI [17, 16]. Five LLMs (GPT-4, GPT-3.5, Davinci-003, Llama, and Galactica) [43] are evaluated for each chemistry task in zero-shot and few-shot in-context learning settings with carefully selected demonstration examples and specific prompts. We highlight the **contributions** of this paper as follows:

- We are the first to establish a comprehensive benchmark to evaluate the abilities of LLMs on a wide range of chemistry tasks. These eight selected tasks, in consultation with chemists, not only encompass a diverse spectrum of the chemistry domain but also demand different abilities such as understanding, reasoning, and explaining using domain-specific chemistry knowledge.

- We provide a comprehensive experimental framework for testing LLMs in chemistry tasks. To factor in the impact of prompts and demonstration examples in in-context learning, we have assessed multiple input options, focusing on the description of chemistry tasks. Five representative configurations were chosen based on their performance on a validation set, then these selected options were applied on the testing set. The conclusion is made from five repeated evaluations on each task, since GPTs often yield different outputs at different API calls even though the input is the same. We thus believe that our benchmarking process is both reliable and systematic.

- Our investigations yield broader insights into the performance of LLMs on chemistry tasks. As summarized in Table 2, our findings confirm some anticipated outcomes (e.g., GPT-4 outperforms GPT-3 and Davinci-003), and also reveal unexpected discoveries (e.g., property prediction can be better solved when property label semantics are included in prompts). Our work also contributes to practical recommendations that can guide AI researchers and chemists in leveraging LLMs more effectively in the future (see Section 5).

The paper is organized as follows. Related works are presented in Section 2. In section 3, we elaborate on the evaluation process, including an overview of the chemistry tasks, the utilized LLMs and prompts, and the validation and testing settings. In section 4, we summarize the main findings (due to the space limit, evaluation details of each chemistry task can be found in Appendix). Finally, to answer the question *"What can LLMs do in chemistry?"* we discuss the constraints inherent to LLMs and how different settings related to LLMs affect performance across various chemistry tasks in Section 5. The conclusions are summarized in section 6.

## 2   Related Work

**Large Language Models.** The rise of Large Language Models (LLMs) has marked a significant trend in recent natural language processing (NLP) research. This progress has been fuelled by milestones such as the introduction of GPT-3 [4], T0 [52], Flan-T5 [12], Galactica [56] and LLaMa [57]. The

recently released GPT-4, an evolution from GPT-3.5 series, has drawn considerable attention for its improvements in language understanding, generation, and planning [43]. Despite the vast potential of LLMs, existing research primarily centers on their performance within general NLP tasks [8, 9]. The scientific disciplines, notably chemistry, have received less focus. The application of LLMs in these specialized domains presents an opportunity for significant advancements. Therefore, we conduct a comprehensive experimental analysis to evaluate the capability of LLMs in chemistry-related tasks.

**Large Language Model Evaluations.** In recent years, the evaluation of LLMs like GPT has become a significant field of inquiry. [11] showed ChatGPT's proficiency in law exams, while technical aspects of GPT-4 were analyzed in [43]. LLMs are also applied in healthcare [14], mathematical problem [18], and code generation tasks [37]. Specifically, in healthcare, the utility and safety of LLMs in clinical settings were explored [42]. In the context of mathematical problem-solving, studies [18, 7] have highlighted that LLMs encounter challenges with graduate-level problems, primarily due to difficulties in parsing complex syntax. These studies underscored the complexity of achieving task-specific accuracy and functionality with LLMs. Lastly, AGIEval [66] assessed LLMs' general abilities but noted struggles in complex reasoning tasks.

Our work aligns with these evaluations but diverges in its focus on chemical tasks. To our knowledge, this is the first study to transform such tasks to suit LLM processing and to perform a comprehensive evaluation of these models' ability to tackle chemistry-related problems. This focus will contribute to expand our understanding of LLMs' capabilities in specific scientific domains.

**Large Language Model for Chemistry.** Recent efforts integrating LLMs with the field of chemistry generally fall into two distinct categories. One category aims to create a chemistry agent with LLMs' by leveraging its planning ability to utilize task-related tools. For example, Bran et al [3] developed ChemCrow, which augmented LLMs with chem-expert designed tools for downstream tasks such as organic synthesis and drug discovery. Similarly, by leveraging the planning and execution ability of multiple LLMs, Boiko et al [2] developed an autonomous chemical agent to conduct chemical experiments. The other category involves direct usage of LLMs for downstream tasks in chemistry [27, 62, 6, 28]. While these studies have explored the performance of LLMs in chemistry-related tasks, a systematic evaluation of their capabilities within this domain has been lacking. Consequently, there is a noticeable gap that calls for a meticulous benchmark to thoroughly assess the potential of LLMs in chemistry. Such a benchmark is crucial not only for identifying the strengths and limitations of these models in a specialized scientific domain, but also to guide future improvements and applications.

# 3 The Evaluation Process and Setting

The evaluation process workflow is depicted in Fig. 1. Guided by co-author Prof. Olaf Wiest (from the Department of Chemistry at the University of Notre Dame), we identify eight tasks in discussion with senior Ph.D. students at the NSF Center for Computer Assisted Synthesis (C-CAS). Following this, we generate, assess, and choose suitable prompts to forward to LLMs. The acquired answers are then evaluated both qualitatively by chemists to identify whether they are helpful in the real-world scenario and quantitatively by selected metrics.

**Chemistry tasks.** In order to explore the abilities of LLMs in the field of chemistry, we concentrate on three fundamental capabilities: understanding, reasoning, and explaining. We examine these competencies through eight diverse and broadly acknowledged practical chemistry tasks. These tasks are summarized in Table 1, in terms of the *task type* from the perspective of machine learning, the *dataset* used for the evaluation, as well as the *evaluation metrics*. The *#ICL candidates* refers to the number of candidate examples, from which we select $k$ demonstration examples, either randomly or based on similarity searches. These candidate sets are the training sets used in classical machine learning models, e.g., in training classifiers or generative models. We set the test set of 100 instances, randomly sampled from the original testing dataset (non-overlapping with the training set). To reduce the influence of the LLMs randomness on the results, each evaluation experiment is repeated five times and the mean and variance are reported.

**LLMs.** For all tasks, we evaluate the performance of five popular LLMs: GPT-4, GPT-3.5 (referred to as GPT-3.5-turbo, also known as ChatGPT), Davinci-003, LLama and Galactica.

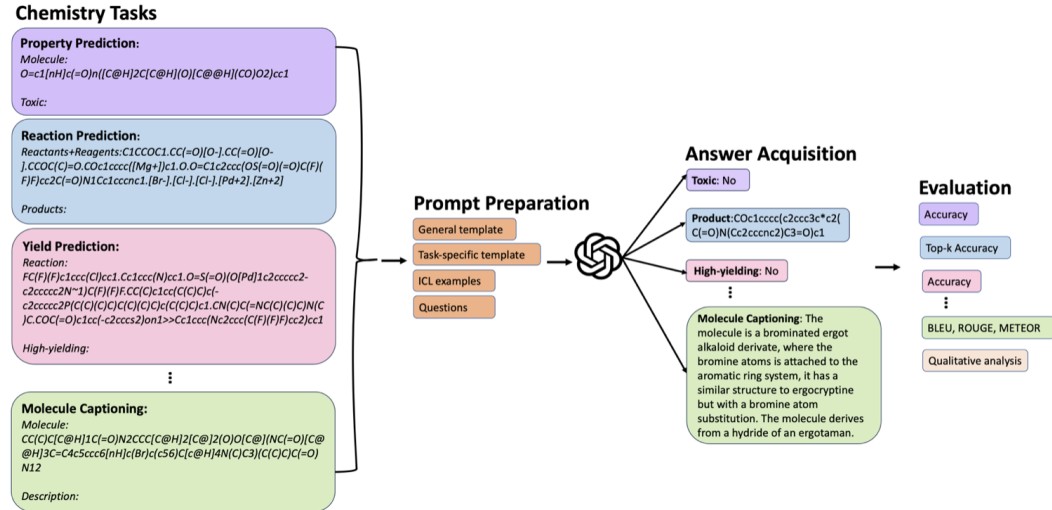

Figure 1: Overview of the evaluation process

Table 1: The statistics of all tasks, datasets, the number of ICL/test samples, and evaluation metrics

| Ability | Task | Task Type | Dataset | #ICL candidates | #test | Evaluation Metrics |
|---------|------|-----------|---------|-----------------|-------|--------------------|
| Understanding | Name Prediction | Generation | PubChem | 500 | 100 | Accuracy |
| | Property Prediction | Classification | BBBP, HIV, BACE, Tox21, ClinTox | 2053, 41127, 1514, 8014, 1484 | 100 | Accuracy, F1 score |
| Reasoning | Yield Prediction | Classification | Buchwald-Hartwig, Suzuki-Miyaura | 3957, 5650 | 100 | Accuracy |
| | Reaction Prediction | Generation | USPTO-Mixed | 409035 | 100 | Accuracy, Validity |
| | Reagents Selection | Ranking | Suzuki-Miyaura | 5760 | 100 | Accuracy |
| | Retrosynthesis | Generation | USPTO-50k | 40029 | 100 | Accuracy, Validity |
| | Text-Based Molecule Design | Generation | ChEBI-20 | 26407 | 100 | BLEU, Exact Match, etc |
| Explaining | Molecule Captioning | Generation | ChEBI-20 | 26407 | 100 | BLEU, Chemists, etc |

**Zero-shot prompt.** For each task, we apply a standardized zero-shot prompt template. As shown in Fig. 2, we instruct the LLMs to act in the capacity of a chemist. The content within the brackets is tailored to each task, adapting to its specific inputs and outputs. The responses from LLMs are confined to only returning the desired output without any explanations.

> *You are an expert chemist. Given the [reactants SMILES / molecular description / …]: [Input], predict the [reaction product SMILES / molecule SMILES / …] using your experienced chemical [reaction prediction / chemical molecule design / …] knowledge. No explanations and other information. Only return the [product SMILES / designed molecular SMILES ].*

Figure 2: The standardized zero-shot prompt template for all tasks.

**Task-specific ICL prompt.** ICL is a new paradigm for LLMs where predictions are based solely on contexts enriched with a few demonstration examples [15]. This paper specifically denotes ICL as a few-shot in-context learning approach, excluding the zero-shot paradigm. In order to thoroughly examine the capacities of LLMs within each chemistry-specific task, we design a task-specific ICL prompt template. As shown in Fig. 3. The format of the template is similar to that used in [48]. We also partition our template into four parts: {General Template}{Task-Specific Template}{ICL}{Question}. The {General Template} is almost the same as the zero-shot prompt, instructing the LLMs to play the role of a chemist and specify the chemistry task with its corresponding input and output. Considering that the responses for chemistry-related tasks must be accurate and chemically reasonable, it is crucial to prevent LLMs from generating hallucinated information. To this end, we introduce the {Task-Specific Template} which consists of three main components: [Input explanation], [Output Explanation], and [Output Restrictions], specifically designed to reduce hallucinations. These components are tailored to each task. The {ICL} part is a straightforward

concatenation of the demonstration examples and it follows the structure "[Input]: [Input_content] [Output]: [Output_content]". The [Input] and [Output] denote the specific names of each task's input and output, respectively. For example, in the reaction prediction task, the [Input] would be "Reactants+Reagents" and the [Input_content] would be the actual SMILES of reactants and reagents. The [Output] would be "Products" and the [Output_content] would be the SMILES of products. Detailed ICL prompts for each task will be presented in their respective sections that follow. The last {Question} part presents the testing case for LLMs to respond to. Fig 5 is example of our name prediction prompt.

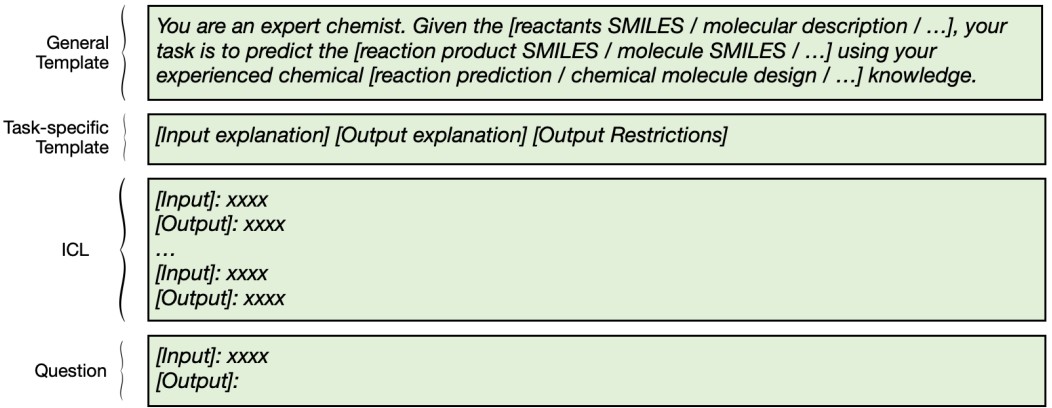

Figure 3: An ICL prompt template for all tasks.

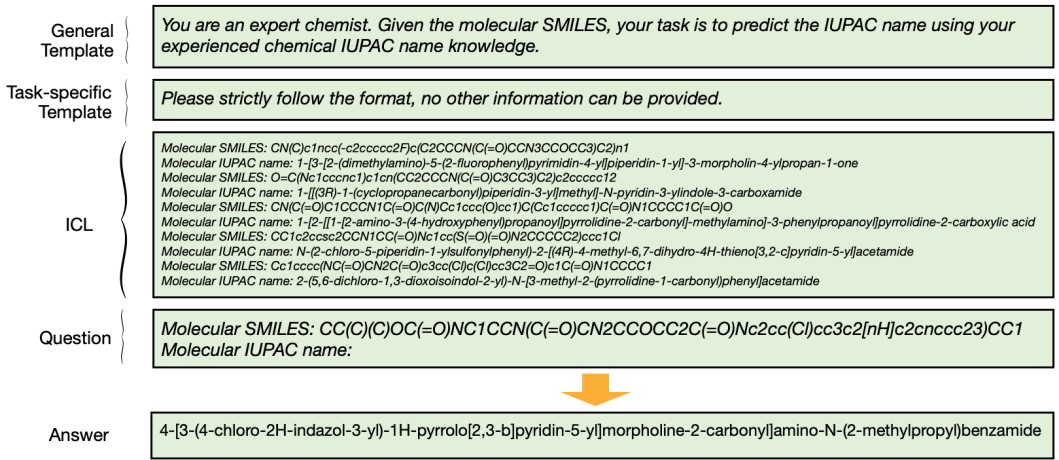

Figure 4: An ICL prompt example for smiles2iupac prediction

**ICL strategies.** To investigate the impact of the quality and quantity of ICL examples on the performance of each task, we explore two ICL strategies. The quality is determined by the retrieval methods employed for finding similar examples to the sample in question. We conduct a grid search across two strategies: {Random, Scaffold}. In the Random strategy, we randomly select $k$ examples from the ICL candidate pool. In the Scaffold strategy, if the [Input_content] is a molecule SMILES, we use Tanimoto Similarity [55] from Morgan Fingerprint [41] with 2048-bits and radius=2 to calculate the molecular scaffold similarity to find the top-$k$ similar molecule SMILES. If the [Input_content] is a description such as IUPAC name or others, we use Python's built-in difflib.SequenceMatcher tool [49] to find the top-$k$ similar strings. To explore the influence of the quantity of ICL examples on performance, we also perform a grid search for $k$, the number of ICL examples, in each task.

**Experiment setup strategy.** In property prediction and yield prediction tasks, we perform the grid search of $k$ in {4, 8}. In the name prediction, reaction prediction, and retrosynthesis tasks, we perform the grid search of $k$ in {5, 20}. In text-based molecule design and molecule captioning tasks, we

Table 2: The rank of five LLMs on eight chemistry tasks and performance highlight (NC: not competitive, C: competitive, SC: selectively competitive, acc: accuracy).

| Task | GPT-4 | GPT-3.5 | Davinci-003 | Llama2-13B-chat | GAL-30B | Performance highlight (comparing to baselines if any) |
|---|---|---|---|---|---|---|
| Name Prediction | 1 | 2 | 3 | 4 | 5 | **NC**: max. acc. 8% (Table 4) |
| Property Prediction | 1 | 2 | 3 | 5 | 4 | **SC**: outperform RF and XGBoost from MoleculeNet [65] (Table 6) |
| Yield Prediction | 1 | 3 | 2 | 5 | 4 | **C**: but 16-20% lower acc. than UAGNN [34] (Table 10) |
| Reaction Prediction | 1 | 3 | 2 | 5 | 4 | **NC**: 70% lower acc. than Chemformer [26] (Table 11) |
| Reagents Selection | 2 | 1 | 3 | 4 | 5 | **C**: 40-50% acc. (Table 12) |
| Retrosynthesis | 2 | 3 | 1 | 5 | 4 | **NC**: 40% lower acc. than Chemformer [26] (Table 13) |
| Molecule Design | 1 | 3 | 2 | 4 | 5 | **SC**: better than MolT5-Large [17] (Table 14) |
| Molecule Captioning | 1 | 2 | 1 | 4 | 5 | **SC**: better than MolT5-Large [17] (Table 15) |
| Average rank | 1.25 | 2.375 | 2.125 | 4.5 | 4.5 | overall: 3 **SC**, 2 **C**, 3 **NC** |

perform the grid search of $k$ in {5, 10} because of the maximum token limitation of LLMs. To reduce the time consumption of API requests caused by testing on the large test set, we first construct a validation set of size 30 which is randomly sampled from the original training set. Then we search $k$ and retrieval strategies ({Random, Scaffold}) on the validation set. Based on the validation set results, we take 5 representative options when testing on 100 instances, which are randomly sampled from the original test set. For each task, we run evaluation 5 times and report mean and standard deviation.

## 4 Experiment Analysis

Due to space limitations, we provide details of the evaluation on each chemistry task in Appendix by the following order: name prediction in section A, property prediction in section B, yield prediction in section C, reaction prediction in section D, reagents selection in section E, retrosynthesis in section F, text-based molecule design in section G, and molecule captioning in section H. The detailed results described in the Appendix allow us to approach the question **"What can LLMs do in chemistry?"** from several directions. We discuss the key findings from our comprehensive benchmark analysis and provide valuable insights by thoroughly analyzing the limitation of LLMs and how different settings related to LLMs affect performance across various chemistry tasks.

### 4.1 Can LLMs outperform existing baselines in chemistry tasks?

Several classic predictive models based on machine learning (ML) have been developed for specific chemistry tasks. For instance, MolR (Graph Neural Network-based) predicts molecule properties as a binary classification problem [58]. UAGNN achieved state-of-the-art performance in yield prediction [34]. MolT5-Large, a specialized language model based on T5, excels in translating between molecule and text [17]. We conduct a performance analysis of GPT models and compare their results with available baselines, if applicable. The **main findings** from the investigations are:

- GPT-4 outperforms the other models evaluated. The ranking of the models on 8 tasks can be found in Table 2;

- GPT models exhibit a less competitive performance in tasks demanding precise understanding of molecular SMILES representation, such as name prediction, reaction prediction and retrosynthesis;

- GPT models demonstrate strong capabilities both qualitatively (in Fig. 14 evaluated by chemists) and quantitatively in text-related explanation tasks such as molecule captioning;

- For chemical problems that can be converted to classification tasks or ranking tasks, such as property prediction, and yield prediction, GPT models can achieve competitive performance compared to baselines that use classical ML models as classifiers, or even better, as summarized in Table 2.

These conclusions are derived from conducting five repeated evaluations on each task, using the best evaluation setting that was discovered through a grid search on the validation set of each task. We designate the performance of GPT models as three categories and provide in-depth discussion next.

- **Tasks with not competitive (NC) performance**. In tasks such as **reaction prediction** and **retrosynthesis**, GPT models are worse than existing ML baselines trained by large amounts of training data, partially because of the limitation on understanding molecular SMILES strings. In **reaction prediction** and **retrosynthesis**, SMILES strings are present in both the input and output of the GPT models. Without an in-depth understanding of the SMILES strings that represent

reactants and products, as well as the reaction process that transforms reactants into products, it will be difficult for GPT models to generate accurate responses, as shown in Table 11 and 13. GPT models exhibit poor performance on the task of **name prediction** as well (see Table 4). This further validates the notion that GPT models struggle with understanding long strings in formats such as SMILES, IUPAC name, and molecular formula, and make correct translations between them.

- **Tasks with competitive (C) performance**. GPT models can achieve satisfactory results when the chemistry tasks are formulated into the forms of classification (e.g., formatting **yield prediction** into a high-or-not classification, instead of regression) or ranking (as seen in **reagents selection**), as illustrated in Fig. 7 and 9. This is understandable, because making choices is inherently simpler than generating products, reactants or names. GPT models can achieve an accuracy of 40% to 50% when asked to select the reactant or solvent or ligand from provided candidates. Although GPT-4's performance on **yield prediction** falls short compared to the baseline model UAGNN [34] (with 80% versus 96% on the Buchwald-Hartwig dataset, and 76% versus 96% on the Suzuki-coupling dataset), it demonstrates improved performance when given more demonstration examples within the few-shot in-context learning scenario, as reported in Table 10. It is worth noting that the UAGNN model was trained on thousands of examples for these specific reactions. Last, while GPT models exhibit promising performance for yield prediction on the evaluated High-Throughput Experimentation (HTE) datasets, specifically the Buchwald-Hartwig [1] and Suzuki-Miyaura datasets [50], they perform as bad as other ML baselines on more challenging datasets like USPTO-50k [53]. This observation indicates a potential area for future research and improvement in the performance of GPT models on challenging chemistry datasets.

- **Tasks with selectively competitive (SC) performance**. GPT models are selectively competitive on two types of tasks.

  - In the **property prediction** task on some datasets (HIV, ClinTox), GPT models outperform the baseline significantly, achieving F1 scores and accuracy nearing 1, as reported in Table 6 and 7. This might be due to the fact that the property labels to be predicted are included in the prompts, with GPT models being simply tasked in responding with *yes* or *no*. For example, the prompt includes *inhibit HIV replication* or *drugs failed clinical trials for toxicity reason*, and we observed a significant decline in the performance of GPT models upon removing property labels from the prompt (refer to Appendix section B). In contrast, baselines employing machine learning models do not include the semantic meaning of these labels in their input. The input for these models only comprises molecular representations in graph form but no labels.

  - For tasks related to text, such as **text-based molecule design and molecule captioning**, GPT models exhibit strong performance due to their language generation capabilities. On the task of **text-based molecule design**, GPT models outperform the baseline when evaluated using NLP metrics such as BLEU and Levenshtein. However, when it comes to exact match, the accuracy is less than 20%, as reported in Table 14 and 15. This suggests that the molecules designed by GPT models may not be exactly the same as the ground truth. Particularly in the context of molecular design/generation, the exact match is a significant metric. Unlike in natural language generation where there is some allowance for deviation from the input, molecular design demands precise accuracy and chemical validity. However, not being precisely identical to the ground truth does not automatically invalidate a result. Molecules generated by GPT models may still prove to be beneficial and could potentially act as viable alternatives to the ground truth, provided they meet the requirements outlined in the input text and the majority (over 89%) are chemically valid (see Table 14). Nonetheless, assessing the true utility of these generated molecules, such as evaluating their novelty in real-world applications, can be a time-consuming undertaking.

## 4.2 The capability of different LLMs

As shown in Table 2, we can find that GPT-4 model shows better chemical understanding, reasoning, and explaining abilities than Davinci-003, GPT-3.5, Llama and Galactica. This further verifies the GPT-4 model outperforms the other models in both basic and realistic scenarios [5].

## 4.3 The effects of the ICL

To investigate the effects of the ICL, we introduced ICL prompting and different ICL retrieval methods, and the different number of ICL examples in each task. Based on the experiments results of

12 different variants of each option and evaluating their performance on the validation set, we have the following three observations:

- In all tasks, the performance of ICL prompting is better than zero-shot prompting.

- In most tasks (in Table 4, 6, 7, 11, 13, 14, 15), using scaffold similarity to retrieve the most similar examples of the question as ICL examples achieves better performance than random sampling.

- In most tasks (in Table 4, 6, 7, 10, 11, 14, 15), using larger $k$ (more ICL examples) usually achieves better performance than small $k$ (fewer ICL examples).

These observations indicate that the quality and quantity of ICL examples plays an important role in the performance of ICL prompting [23, 36]. This may inspire that it is necessary to design more chemistry-specific ICL methods to build high-quality ICL examples to further improve the ICL prompting performance.

## 4.4 Are molecule SELFIES representations more suitable for LLMs than SMILES representations?

SELFIES [33] representations are more machine-learning-friendly string representations of molecules. To investigate whether the SELFIES representations are more suitable for LLMs than SMILES representations, we conduct experiments on four tasks, including molecule property prediction, reaction prediction, molecule design and molecule captioning. The experiment results are shown in Table 16, 17, 18, 19. We can observe that the **results of using SELFIES in all four tasks are inferior to those of using SMILES**. This could be attributed to the fact that the pretraining datasets for LLMs are primarily populated with SMILES-related content rather than SELFIES. Consequently, these models are more attuned to SMILES. However, it's worth mentioning that the occurrence of invalid SELFIES is less frequent than that of invalid SMILES, which aligns with the inherent design of SELFIES to ensure molecular validity.

## 4.5 The impact of temperature parameter of LLMs

One key hyperparameter that affects the performance of LLMs is temperature, which influences the randomness in the model's predictions. To determine the optimal temperature for each task, we randomly sampled 30 data points from the datasets and performed in-context learning experiments across various temperature settings. While optimal temperatures determined on the validation set may not always yield optimal results on the test set, our methodology is primarily designed to conserve token usage and API query time. To address potential discrepancies between validation and test sets, we performed targeted temperature testing on the test sets for two molecular property prediction datasets: BBBP and BACE. Our results are summarized in Table 3. For these tests, we employed the GPT-4 model (using scaffold sampling with $k = 8$) and set temperature values $t = [0.2, 0.4, 0.6, 0.8, 1]$. The result reveal that variations in the temperature parameter have a marginal impact on test performance, with fluctuations of less than $0.05$ observed in both F1 and accuracy scores. These results validate the robustness of our initial sampling approach and underscore the reliability of our findings across different settings.

Table 3: The F1($\uparrow$) and accuracy($\uparrow$) score of GPT-4 model(scaffold sampling, $k = 8$) on different temperature setting.

| F1($\uparrow$) | BBBP | BACE | Accuracy($\uparrow$) | BBBP | BACE |
|---|---|---|---|---|---|
| GPT-4(t=0.2) | $0.667 \pm 0.029$ | $0.741 \pm 0.019$ | GPT-4(t=0.2) | $0.650 \pm 0.028$ | $0.743 \pm 0.019$ |
| GPT-4(t=0.4) | $0.712 \pm 0.014$ | $0.728 \pm 0.024$ | GPT-4(t=0.4) | $0.691 \pm 0.017$ | $0.729 \pm 0.024$ |
| GPT-4(t=0.6) | $0.683 \pm 0.016$ | $0.736 \pm 0.020$ | GPT-4(t=0.6) | $0.659 \pm 0.016$ | $0.736 \pm 0.019$ |
| GPT-4(t=0.8) | $0.686 \pm 0.030$ | $0.744 \pm 0.025$ | GPT-4(t=0.8) | $0.661 \pm 0.032$ | $0.745 \pm 0.025$ |
| GPT-4(t=1.0) | $0.684 \pm 0.023$ | $0.756 \pm 0.025$ | GPT-4(t=1.0) | $0.660 \pm 0.021$ | $0.757 \pm 0.025$ |

# 5 Discussion

## 5.1 Limitation of LLMs on understanding molecular SMILES

A significant limitation of LLMs is their lack of understanding of molecular representations in SMILES strings, which in many cases leads to inaccurate or inconsistent results as shown in Section A for the translation of different ways to name molecules. SMILES (Simplified Molecular Input Line Entry System) [60, 61] is a widely used textual representation for chemical structures. For example, the SMILES string for ethanol, a simple alcohol, is "CCO". This string represents a molecule with two carbon atoms (C) connected by a single bond and an oxygen atom (O) connected to the second carbon atom. SMILES strings can serve as both input and output for LLMs, alongside other natural language text. However, several issues make it challenging for LLMs to accurately understand and interpret SMILES strings: **1)** Hydrogen atoms are not explicitly represented in SMILES strings, as they can be inferred based on the standard bonding rules. LLMs frequently struggle to infer these implicit hydrogen atoms and may even fail at simple tasks like counting the number of atoms in a molecule [27, 6]. **2)** A given molecule can have multiple valid SMILES representations, which can lead to ambiguity if not properly processed or standardized. LLMs may thus fail to consistently recognize and compare molecular structures represented by different SMILES strings. **3)** LLMs do not have any inherent understanding of SMILES strings, and treat them as a sequence of characters or subwords. When processing long SMILES strings, LLMs rely on the byte-pair encoding tokenization technique, which can break the string into smaller pieces or subwords in ways that do not represent the molecular structure and properties of molecules represented by SMILES strings. Because many tasks in cheminformatics rely on the accurate representation of a molecule by SMILES strings, the non-competitive performance of GPT models in converting structures into SMILES strings (and vice versa) affects downstream tasks such as retrosynthesis, reaction and name prediction. LLMs that have an enhanced ability of handling molecular structures and their specific attributes or coupling to existing tools such as RDKit [35] will be needed.

## 5.2 The limitations of current evaluation methods

Although in Text-Based Molecule Design and Molecule Captioning tasks, GPT models show competitive performance compared to the baseline in some metrics (BLEU, Levenshtein, ROUGE, FCD, etc), we observe that the exact match of GPT models is inferior to the baseline in the Text-Based Molecule Design task and the GPT models generate some descriptions which violate chemical facts. This divergence between metrics and real-world scenarios mainly arises because, unlike many natural language processing tasks that can be suitably evaluated by sentence-level matching evaluation metrics, chemistry-related tasks necessitate exact matching for SMILES and precise terminology in descriptions. These findings spotlight the limitations of current evaluation metrics and underscore the need for the development of chemistry-specific metrics.

## 5.3 Hallucination of LLMs in chemistry

Our evaluation experiments across various tasks reveal two primary types of hallucinations exhibited by LLMs in the domain of chemistry. The first type occurs when the input is given in SMILES format (e.g., name prediction); LLMs occasionally struggle with interpreting these SMILES correctly. For instance, they may fail to recognize the number of atoms or certain functional groups within molecules during name prediction tasks. The second type of hallucination arises when the expected output from LLMs should be in the form of SMILES (e.g., reaction prediction and retrosynthesis). Here, LLMs may produce molecules that are chemically unreasonable, suggesting a gap in understanding what constitutes valid SMILES. Hallucination issues represent a key challenge with LLMs, particularly in the field of chemistry which necessitates exact matching of SMILES and adherence to strict chemical facts [62]. Current LLMs need further investigation into this problem.

## 5.4 Prospects of LLMs for chemistry

Overall, through an exhaustive set of experiments and analyses, we outline several promising avenues for the application of LLMs in the field of chemistry. While LLMs underperform relative to baselines across a majority of tasks, it's important to note that LLMs leverage only a few examples to solve chemistry problems, whereas baselines are trained on extensive, task-specific datasets and are limited

to certain tasks. This observation provides valuable insights into the potential of LLMs' generalized intelligence in the domain of chemistry. The employment of advanced prompting techniques such as Chain-of-thought (CoT) [59], Decomposed Prompting [31] could potentially boost the capacity of LLMs to perform complex reasoning. On the other hand, LLMs display a considerable amount of hallucinations in chemistry tasks, indicating that current LLMs may not yet possess the necessary capabilities to solve practical chemistry problems effectively. However, with continuous development of LLMs and further research into methods to avoid hallucinations, we are optimistic that LLMs can significantly enhance their problem-solving abilities in the field of chemistry.

### 5.5 Impact of generating harmful chemicals

Our work demonstrate that LLMs can generate chemically valid molecules. However, it's crucial to acknowledge and mitigate the risks of AI misuse, such as generating hazardous substances. While advancements in AI-enabled chemistry have the potential to bring about groundbreaking medicines and sustainable materials, the same technology can be misused to create toxic or illegal substances. This dual-edged potential emphasizes the necessity for stringent oversight. Without careful regulation, these tools could not only pose significant health and safety hazards but also create geopolitical and security challenges. Consequently, as we harness the capabilities of LLMs in the field of chemistry, we concur with earlier research on generative models in chemistry [2, 3] that it is vital for developers to establish robust safeguards and ethical guidelines to deter harmful applications. This is akin to the limitations imposed on popular search engines, which can also be exploited to find information about dangerous chemicals or procedures online.

### 5.6 Broader Impacts

Our work has broad impacts across multiple dimensions. First, it offers valuable insights and recommendations for both AI researchers and chemists in academia and industry. These perspectives enhance the effective utilization of LLMs and guide future advancements in the field. Second, our objective evaluation of LLMs helps alleviate concerns regarding the replacement of chemists by AI. This aspect contributes to public education, addressing misconceptions and fostering a better understanding of the role of AI in chemistry. Furthermore, we provide a comprehensive experimental framework for testing LLMs in chemistry tasks, which can also be applicable to other domains. This framework serves as a valuable resource for researchers seeking to evaluate LLMs in diverse fields. However, it is important to recognize the ethical and societal implications associated with our work. Additionally, concerns about job displacement in the chemical industry may arise, and efforts should be made to address these challenges and ensure a responsible and equitable adoption of AI technologies.

## 6 Conclusion and Future Work

In this paper, we summarize the required abilities of LLMs in chemistry and construct a comprehensive benchmark to evaluate the five most popular LLMs (GPT-4, GPT-3.5, Davinci-003, LLama and Galactica) on eight widely-used chemistry tasks. The experiment results show that LLMs perform less competitive in generative tasks which require in-depth understanding of molecular SMILES strings, such as reaction prediction, name prediction, and retrosynthesis. LLMs show competitive performance in tasks that are in classification or ranking formats such as yield prediction and reagents selection. LLMs are selectively competitive on tasks involving text in prompts such as property prediction and text-based molecule design, or explainable tasks such as molecule captioning. These experiments indicate the potential of LLMs in chemistry tasks and the need for further improvement. We will collaborate with more chemists in the C-CAS group, progressively integrating a wider range of tasks that are both novel and practical. We hope our work can address the gap between LLMs and the chemistry research field, inspiring future research to explore the potential of LLMs in chemistry.

## Acknowledgments and Disclosure of Funding

This work was supported by the National Science Foundation (CHE–2202693) through the NSF Center for Computer Assisted Synthesis (C-CAS).

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

# Appendix

## A   Name Prediction

For one molecule, there are different chemical naming conventions and representations such as SMILES, IUPAC names, and graphic molecular formula. To investigate whether GPT models have the basic chemical name understanding ability, we construct four chemical name prediction tasks that include SMILES to IUPAC name translation (smiles2iupac), IUPAC name to SMILES translation (iupac2smiles), SMILES to molecule formula translation (smiles2formula), and IUPAC name to molecule formula translation (iupac2formula). We collect 630 molecules and their corresponding names including SMILES, IUPAC name, and molecule formula from PubChem[3] [32]. We randomly sample 500 molecules as the ICL candidates, and other 30 molecules as the validation set, and other 100 molecules as the test set. For all name translation tasks, we use the exact match accuracy as the metric to evaluate the performance.

**ICL Prompt.**   One example of the smiles2iupac prediction is shown in Figure 5. For other name translation tasks, we only change the underlined parts that represent different tasks and their corresponding input names and output names.

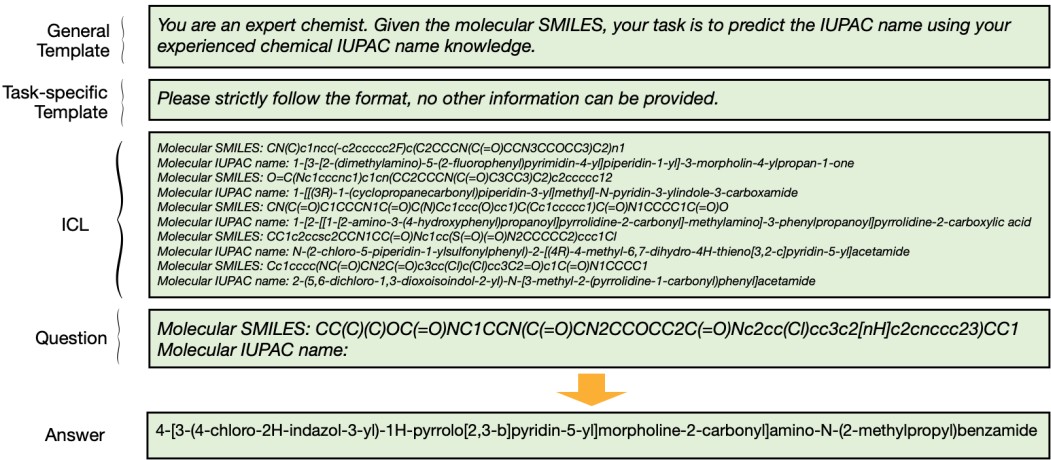

Figure 5: An ICL prompt example for smiles2iupac prediction

**Results.**   The results are reported in Table 4 (we only report representative methods along with their optimal prompt settings via grid search on validation set). In all four name prediction tasks, the accuracy of the best method is extremely low (0.014 in the iupac2smiles task, 0.086 in the smiles2formula task, 0.118 in the iupac2formula task) or even 0 (in the smiles2iupac task). This indicates the LLMs lack basic chemical name understanding ability. The accuracy of Davinci-003 is considerably inferior to other models.

**Case studies.**   Example results generated by GPT-4 (Scaffold, $k$=20) method for each task is shown in Table 5. In all tasks, the GPT-4 model gives the wrong answers. In the smiles2formula task, we can observe that GPT models cannot even recognize the number of Carbon and infer the correct number of Hydrogen, demonstrating the bad chemical understanding ability of GPT models. For prospects, some pre-training technologies such as wrapping molecules with text [38] or code-switch [64, 20] may be helpful to align different chemical names of the same molecule to help improve LLMs' chemical understanding.

---

[3]https://pubchem.ncbi.nlm.nih.gov

Table 4: The accuracy (↑) of LLMs in 4 different name prediction tasks. The best LLM is in bold font. Here $k$ is the number of examples used in few-shot ICL. The baseline is underlined and "-" indicates that STOUT cannot solve the smiles2formula and iupac2formula tasks.

| Method | smiles2iupac | iupac2smiles | smiles2formula | iupac2formula |
|---|---|---|---|---|
| STOUT [47] | 0.55 | 0.7 | - | - |
| GPT-4 (zero-shot) | 0 | 0.008±0.008 | 0.048± 0.022 | 0.092±0.018 |
| GPT-4 (Scaffold, $k$=5) | 0 | **0.014±0.009** | 0.058±0.015 | **0.118±0.022** |
| GPT-4 (Scaffold, $k$=20) | 0 | 0.012±0.004 | **0.086±0.036** | 0.084±0.005 |
| GPT-4 (Random, $k$=20) | 0 | 0.010±0.007 | 0.070±0.032 | 0.076±0.011 |
| GPT-3.5 (Scaffold, $k$=20) | 0 | 0.010±0.000 | 0.052±0.004 | 0.044±0.009 |
| Davinci-003 (Scaffold, $k$=20) | 0 | 0 | 0.006±0.005 | 0.018±0.004 |
| Llama2-13B-chat (Scaffold, $k$=20) | 0 | 0 | 0.010±0.007 | 0 |
| GAL-30B (Scaffold, $k$=10) | 0 | 0 | 0 | 0 |

Table 5: Example results generated by GPT-4 (Scaffold, $k$=20) method for different tasks

| Task | Input | Ground Truth | Output of GPT-4 (Scaffold, k=20) |
|---|---|---|---|
| smiles2iupac | CCOC(=O)C(C(C)=O)=C(C)N | ethyl 2-acetyl-3-aminobut-2-enoate | ethyl 2-methyl-5-oxo-2-azahept-4-en-3-oate |
| iupac2smiles | ethyl 2-acetyl-3-aminobut-2-enoate | CCOC(=O)C(C(C)=O)=C(C)N | CCOC(=O)C=C(C)C(=N)C |
| smiles2formula | Cc1noc(CCn2cc[nH]c2=O)n1 | C8H10N4O2 | C9H10N4O2 |
| iupac2formula | R)-(1-benzylquinolin-1-ium-4-yl) -(5-ethenyl-1-azabicyclo[2.2.2]octan-2-yl)methanol;chloride | C26H29ClN2O | C23H27ClN2O |

# B    Molecule Property Prediction

Molecule property prediction [21, 58] is a fundamental task in computational chemistry that has been gaining significant attention in recent years due to its potential for drug discovery, material science, and other areas in the chemistry. The task involves using machine learning techniques [22] to predict the chemical and physical properties of a given molecule, based on its molecular structure. We aim to further explore the potential of LLMs in molecular property prediction and assess their performance on a set of benchmark datasets, such as BBBP(MIT license), HIV(MIT license), BACE(MIT license), Tox21(MIT license), and ClinTox(MIT license), which were originally introduced by [65]. The datasets are made up of extensive collections of SMILES, paired with binary labels that highlight the particular property being evaluated, such as BBBP: Blood-Brain Barrier Penetration, HIV: inhibit HIV replication, BACE: bindings results for a set of inhibitors of human beta-secretase, Tox21: toxicity of compounds, and ClinTox: drugs failed clinical trials for toxicity reasons. A comprehensive explanation of these datasets can be referenced in the original research conducted by [65]. For ICL, we either select $k$ samples randomly, or search the top-$k$ most analogous molecules using RDKit [35] to determine the Tanimoto Similarity. However, it is crucial to mention that using the latter method does not assure an even distribution among classes. In our study, we employ a strategic sampling method for two categories of datasets: balanced and highly imbalanced. For balanced datasets, such as BBBP and BACE, we randomly select 30 samples for the validation process and 100 samples for testing from the original dataset. Contrastingly, for datasets exhibiting substantial label imbalance ($39684:1443 \approx 28:1$, take HIV datasets as a example), we select samples from the majority and minority classes to achieve a ratio of 4:1. This strategic approach enables us to maintain a representative sample for the evaluation process, despite the original high imbalance in the dataset. To evaluate the results, we use the classification *accuracy*, as well as *F1* score as the evaluation metric due to the class imbalance. We benchmark our method against two established baselines from MoleculeNet [65]: RF and XGBoost. Both baselines utilize the 1024-bit circular fingerprint as input to predict the property as a binary classification problem.

**ICL Prompt.**    Figure 6 illustrates a sample of our ICL prompt for property prediction. Within the task-specific template, we include a detailed explanation of the task forecasting the penetration of the brain-blood barrier to assist LLMs in comprehending the input SMILES from the BBBP dataset. Additionally, we establish certain constraints for the output to conform to the specific characteristics of the property prediction task.

**Results.**    The results are reported as F1 in Table 6, accuracy in Table 7. We observed that GPT models outperform the baseline model in terms of F1 on four out of five datasets. In the range of GPT

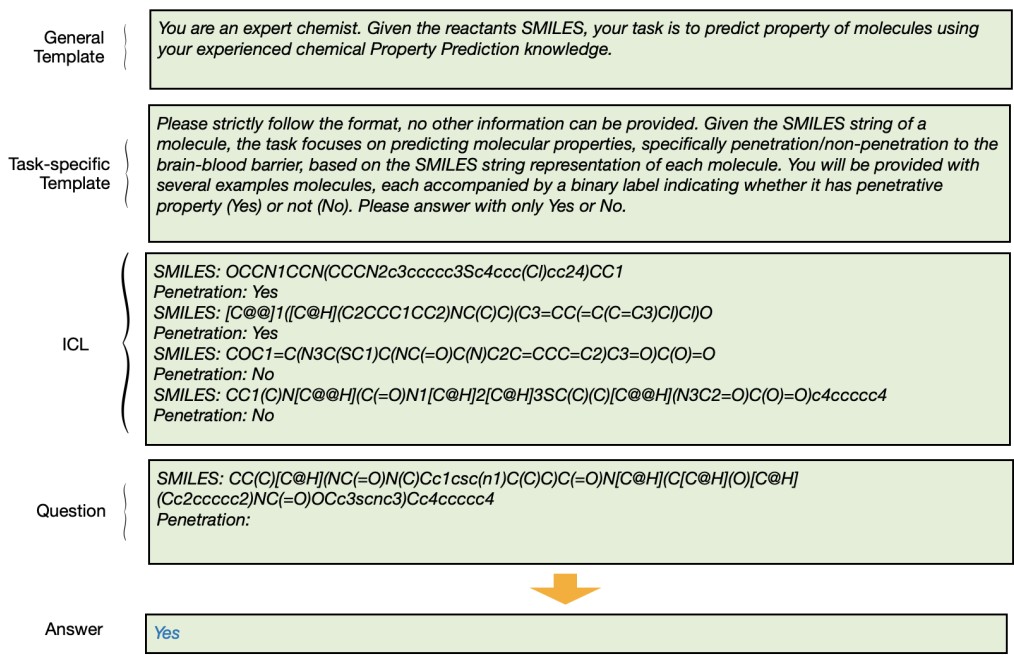

Figure 6: An ICL prompt example for property prediction

Table 6: F1 (↑) score of LLMs and baseline in molecular property prediction tasks. $k$ is the number of examples used in few-shot ICL. The best GPT model is in bold font, and the baseline is underlined.

|  | BBBP | BACE | HIV | Tox21 | ClinTox |
|---|---|---|---|---|---|
| RF | 0.881 | 0.758 | 0.518 | 0.260 | 0.461 |
| XGBoost | 0.897 | 0.765 | 0.551 | 0.333 | 0.620 |
| GPT-4 (zero-shot) | 0.560 ±0.034 | 0.322±0.018 | 0.977±0.013 | 0.489±0.018 | 0.555±0.043 |
| GPT-4 (Scaffold, $k=4$) | 0.498 ± 0.028 | 0.516 ± 0.024 | 0.818 ± 0.015 | 0.444 ± 0.004 | 0.731 ± 0.035 |
| GPT-4 (Scaffold, $k=8$) | **0.587±0.018** | **0.666±0.023** | 0.797 ± 0.021 | **0.563±0.008** | 0.736±0.033 |
| GPT-4 (random, $k=8$) | 0.469 ± 0.025 | 0.504 ± 0.020 | **0.994 ± 0.006** | 0.528±0.003 | **0.924±0.000** |
| GPT-3.5 (Scaffold, $k=8$) | 0.463 ± 0.008 | 0.406 ± 0.011 | 0.807 ± 0.021 | 0.529 ± 0.021 | 0.369 ± 0.029 |
| Davinci-003 (Scaffold, $k=8$) | 0.378 ± 0.024 | 0.649 ± 0.021 | 0.832 ± 0.020 | 0.518±0.009 | 0.850 ± 0.020 |
| Llama2-13B-chat (Scaffold, $k=8$) | 0.002 ± 0.001 | 0.045 ± 0.015 | 0.069 ± 0.033 | 0.047 ± 0.013 | 0.001 ± 0.003 |
| GAL-30B (Scaffold, $k=8$) | 0.074 ± 0.019 | 0.025 ± 0.013 | 0.014 ± 0.016 | 0.077 ± 0.046 | 0.081 ± 0.015 |

Table 7: Accuracy (↑) of LLMs and baseline in molecular property prediction tasks. $k$ is the number of examples used in few-shot ICL. The best GPT model is in bold font, and the baseline is underlined.

|  | BBBP | BACE | HIV | Tox21 | ClinTox |
|---|---|---|---|---|---|
| RF | 0.820 | 0.790 | 0.870 | 0.830 | 0.858 |
| XGBoost | 0.850 | 0.810 | 0.870 | 0.840 | 0.888 |
| GPT-4 (zero-shot) | 0.476 ± 0.036 | 0.499 ± 0.005 | 0.986 ± 0.007 | 0.518 ± 0.018 | 0.736 ± 0.027 |
| GPT-4 (Scaffold, $k=4$) | 0.516 ± 0.022 | 0.514 ± 0.205 | 0.834 ± 0.014 | 0.457 ± 0.004 | 0.856 ± 0.014 |
| GPT-4 (Scaffold, $k=8$) | **0.614±0.016** | **0.679±0.205** | 0.836 ± 0.020 | 0.737 ± 0.004 | 0.856 ± 0.014 |
| GPT-4 (random, $k=8$) | 0.610 ± 0.021 | 0.588 ± 0.023 | **0.996±0.004** | **0.874±0.003** | **0.930±0.010** |
| GPT-3.5 (Scaffold, $k=8$) | 0.463 ± 0.007 | 0.496 ± 0.016 | 0.864 ± 0.014 | 0.572 ± 0.026 | 0.578 ± 0.029 |
| Davinci-003 (Scaffold, $k=8$) | 0.396 ± 0.023 | 0.650 ± 0.021 | 0.781 ± 0.004 | 0.682 ± 0.006 | 0.845 ± 0.010 |
| Llama2-13B-chat (Scaffold, $k=8$) | 0.002 ± 0.003 | 0.048 ± 0.017 | 0.048 ± 0.025 | 0.053 ± 0.011 | 0.002 ± 0.004 |
| GAL-30B (Scaffold, $k=8$) | 0.062 ± 0.007 | 0.020 ± 0.010 | 0.012 ± 0.009 | 0.030 ± 0.018 | 0.099 ± 0.007 |

models examined, GPT-4 surpasses both Davinci-003 and GPT-3.5 in predicting molecular properties. In our investigation, we have found evidence to support that the expansion of in-context learning (ICL) instances leads to a measurable enhancement in model performance. This underlines a direct relationship between the extent of ICL data and the predictive precision of our models. Concurrently, our research presents empirical evidence that scaffold sampling exceeds the performance of random

sampling on three distinct datasets (BBBP, BACE, Tox21). A plausible explanation for this could be the structural resemblances between the scaffold-sampled molecules and the query molecule, which potentially biases the GPT models towards more accurate decision.

**Label interpretation.** The results presented in Table 6 and Table 7 indicate that the GPT-4 model selectively outperforms the baseline models on the HIV and ClinTox datasets. This superior performance likely stems from the inclusion of information directly related to the labels within the ICL prompts. Specifically, in the HIV dataset, the activity test results play a crucial role. Molecules tend to inhibit HIV replication when the activity test is categorized as "confirmed active" or "confirmed moderately active." For the ClinTox dataset, the FDA-approval status of a molecule acts as a predictor of its clinical toxicity. A molecule not having FDA approval is more likely to be clinically toxic. In experiments where we excluded this contextual information from the in-context learning prompts, the F1 and accuracy score of predictions notably declined, as evident from the results in Table 8 and Table 9.

Table 8: Impact to F1 score of removing label context information from the in-context learning prompts.

| F1($\uparrow$) | HIV | ClinTox |
| --- | --- | --- |
| GPT-4(zero-shot) | $0.977 \pm (0.013)$ | $0.489 \pm (0.018)$ |
| GPT-4(unlabelled, zero-shot) | $0.554 \pm (0.017)$ | $0.438 \pm (0.045)$ |
| GPT-4(few-shot) | $0.797 \pm (0.021)$ | $0.563 \pm (0.008)$ |
| GPT-4(unlabelled, few-shot) | $0.493 \pm (0.030)$ | $0.478 \pm (0.035)$ |

Table 9: Impact to accuracy of removing label context information from the in-context learning prompts.

| Accuracy($\uparrow$) | HIV | ClinTox |
| --- | --- | --- |
| GPT-4(zero-shot) | $0.986 \pm (0.070)$ | $0.736 \pm (0.027)$ |
| GPT-4(unlabelled, zero-shot) | $0.628 \pm (0.016)$ | $0.602 \pm (0.039)$ |
| GPT-4(few-shot) | $0.836 \pm (0.020)$ | $0.856 \pm (0.014)$ |
| GPT-4(unlabelled, few-shot) | $0.541 \pm (0.032)$ | $0.630 \pm (0.014)$ |

## C Yield Prediction

Yield prediction [51] is a critical task in chemistry, specifically in the domain of synthetic chemistry, which involves the design and synthesis of new compounds for various applications, such as pharmaceuticals, materials, and catalysts. The yield prediction task aims to estimate the efficiency and effectiveness of a chemical reaction, primarily by quantifying the percentage of the desired product formed from the reactants. We use two High-Throughput experimentation (HTE) datasets: Buchwald-Hartwig [1] (MIT license) and Suzuki-Miyaura dataset [50] (MIT license) for evaluation. These datasets consist of reactions and their corresponding yields, which have been meticulously acquired through standardized and consistent experimental setups. This uniformity ensures that the data within each dataset is coherent, reducing the likelihood of discrepancies arising from variations in experimental procedures or conditions. We formulate the task of yield prediction as a binary classification problem, by determining whether a reaction is a high-yielding reaction or not. We used only random sampling for our ICL examples as reactions in those datasets belong to the same type. For every dataset, we randomly select 30 samples for the validation process and 100 samples for testing from the original dataset. To evaluate the results, we use the classification accuracy as the evaluation metric, with UAGNN [34] serving as baseline. UAGNN reports state-of-the-art performance on yield prediction. It takes the graphs of reactants and products as input, and learns representation of these molecules through a graph neural network, and then predicts the scaled yield .

**ICL prompt.** We show our ICL prompt for yield prediction with an example from Buchwald-Hartwig dataset. As described in Figure 7, we incorporate an input explanation (wherein the reactants are separated by '.' and the products are split by '>>') to assist large language models. Additionally, output restrictions are enforced to ensure the generation of valid results.

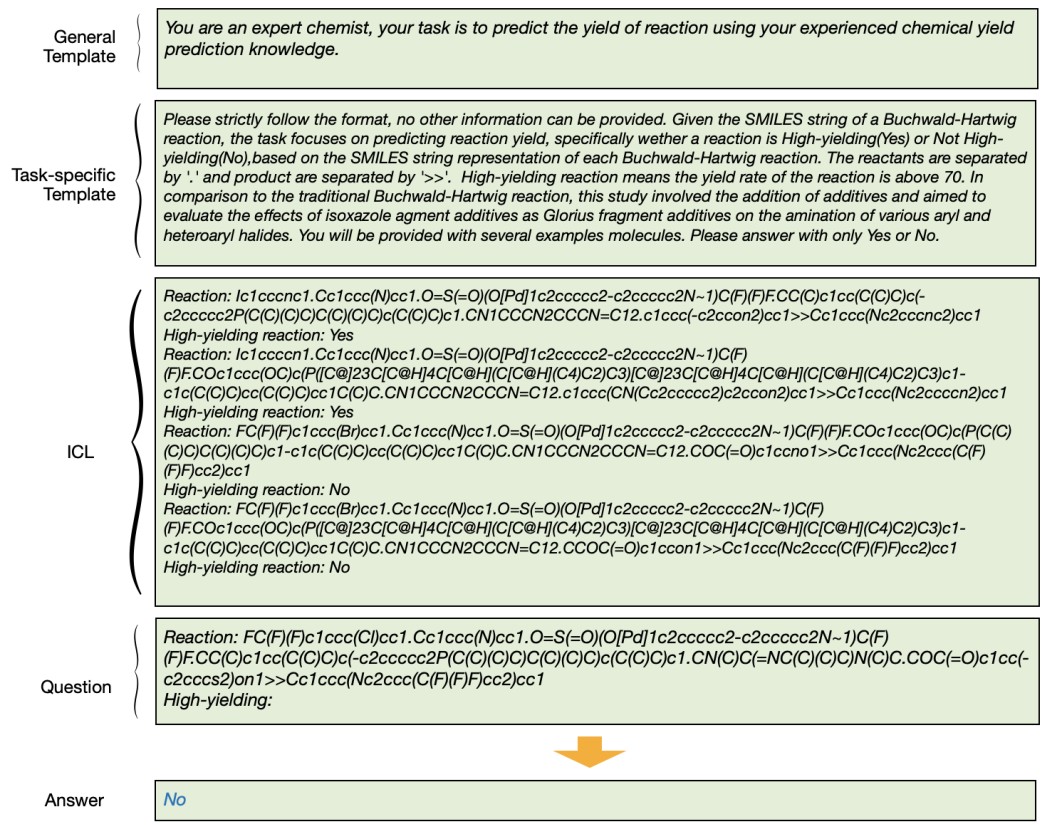

Figure 7: An ICL prompt example for yield prediction

**Results.** The results are presented in Table 10. Our analysis reveals that in the task of yield prediction, GPT models perform below the established baseline model, UAGNN. However, it's worth noting that the UAGNN model was trained on the full training dataset including thousands of examples. Considering the spectrum of GPT models under scrutiny, GPT-4 emerges as the superior model, overshadowing both Davinci-003 and GPT-3.5 in predicting reaction yields. In the process of our investigation, we unearthed supporting evidence that signifies the role of ICL instances in the enhancement of model performance. This suggests an inherent correlation between the quantity of ICL data and the predictive accuracy of the models under consideration. This phenomenon is particularly in the case of GPT-4, we observed a significant improvement in performance when the number of ICL examples was increased from 4 to 8, both in the Buchwald-Hartwig and Suzuki-coupling reactions. This indicates that even within the same model architecture, the amount of contextual data can significantly influence the predictive capabilities.

Table 10: Accuracy ($\uparrow$) of yield prediction task. $k$ is the number of examples used in few-shot ICL. The best LLM is in bold font, and the baseline is underlined.

|  | Buchwald-Hartwig | Suzuki-coupling |
|---|---|---|
| UAGNN [34] | 0.965 | 0.957 |
| GPT-4 (zero-shot) | $0.322 \pm 0.034$ | $0.214 \pm 0.019$ |
| GPT-4 (random, $k$= 8) | **0.800±0.008** | **0.764±0.013** |
| GPT-4 (random, $k$= 4) | $0.574 \pm 0.045$ | $0.324 \pm 0.018$ |
| GPT-3.5 (random, $k$= 8) | $0.585 \pm 0.045$ | $0.542 \pm 0.011$ |
| Davinci-003 (random, $k$= 8) | $0.467 \pm 0.013$ | $0.341 \pm 0.017$ |
| Llama2-13B-chat | $0.008 \pm 0.007$ | $0.006 \pm 0.004$ |
| GAL-30B | 0 | $0.008 \pm 0.010$ |

# D  Reaction Prediction

Reaction prediction is a central task in the field of chemistry, with significant implications for drug discovery, materials science, and the development of novel synthetic routes. Given a set of reactants, the goal of this task is to predict the most likely products formed during a chemical reaction [54, 13, 19]. In this task, we use the widely adopted USPTO-MIT dataset [29](MIT license) to evaluate the performance of GPT models. This dataset contains approximately 470,000 chemical reactions extracted from US patents. In the experiment, we used the USPTO mixed data set, where the reactants and reagents strings are split by '.'. We randomly sampled 30 samples from the original validation set for validation and 100 samples from the original test set for testing. We use the Top-1 Accuracy as the evaluation metric and Chemformer [26] as the baseline due to its superior performance among the machine learning solutions for reaction prediction. Chemformer is a seq2seq model trained to predict the output product when given reactants and reagents as input. We also report the percentage of invalid SMILES generated by each method.

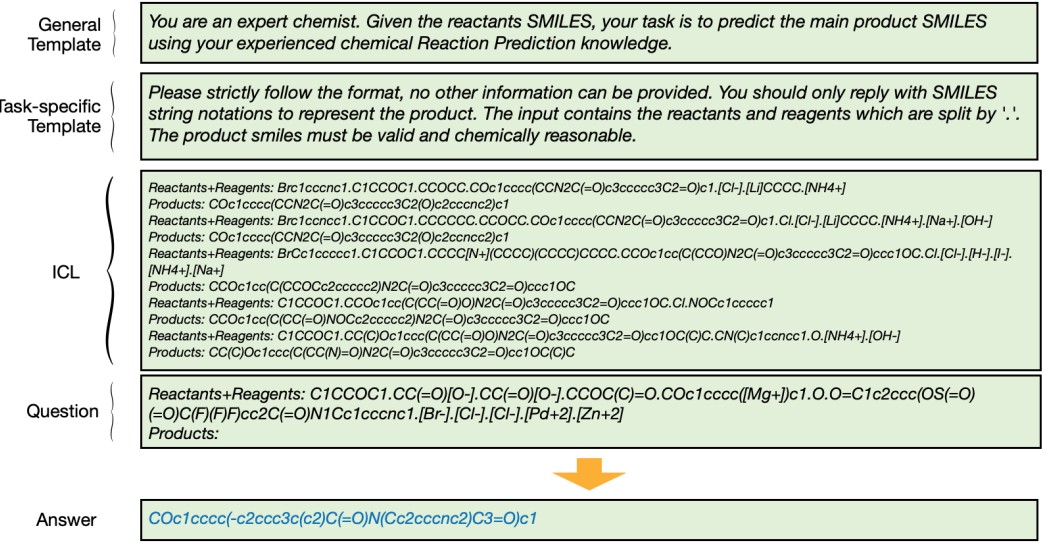

Figure 8: An ICL prompt example for reaction prediction

**ICL Prompt.**  One example of our ICL prompt for reaction prediction is shown in Figure 8. Given the nature of the reaction prediction task and the characteristics of the USPTO-MIT dataset, we enhance the task-specific template with an input explanation (stating that the input includes reactants and reagents, which are separated by '.') to assist the GPT models in understanding the input SMILES. Moreover, we incorporate output restrictions to guide GPT models in generating chemically valid and reasonable products.

Table 11: The performance of LLMs and baseline in the reaction prediction task. $k$ is the number of examples used in few-shot ICL. The best LLM is in bold font, and the baseline is underlined.

| Method | Top-1 Accuracy (↑) | Invalid SMILES (↓) |
|---|---|---|
| Chemformer [26] | 0.938 | 0% |
| GPT-4 (zero-shot) | $0.004 \pm 0.005$ | $17.4\% \pm 3.9\%$ |
| GPT-4 (Scaffold, $k$=20) | $\mathbf{0.230 \pm 0.022}$ | $7.0\% \pm 1.6\%$ |
| GPT-4 (Random, $k$=20) | $0.012 \pm 0.008$ | $8.4\% \pm 1.5\%$ |
| GPT-4 (Scaffold, $k$=5) | $0.182 \pm 0.015$ | $6.6\% \pm 1.5\%$ |
| GPT-3.5 (Scaffold, $k$=20) | $0.184 \pm 0.005$ | $15.6\% \pm 2.3\%$ |
| Davinci-003 (Scaffold, $k$=20) | $0.218 \pm 0.008$ | $11.4\% \pm 2.7\%$ |
| Llama2-13B-chat (Scaffold, k=20) | $0.032 \pm 0.013$ | $27.8\% \pm 5.5\%$ |
| GAL-30B (Scaffold, k=5) | $0.036 \pm 0.011$ | $\mathbf{5.2\% \pm 1.5\%}$ |

**Results.** The results are reported in Table 11. We can observe that compared to the baseline, the performance of GPT models is considerably inferior, especially for the Zero-shot prompting (Top-1 Accuracy is only 0.004 and it generates 17.4% invalid SMILES). The less competitive results of GPT models can be attributed to the lack of in-depth understanding of the SMILES strings that represent reactants and products, as well as the reaction process that transforms reactants into products. It is also worth mentioning that the high accuracy achieved by Chemformer is due to its training on the complete dataset. More conclusions and detailed analysis are summarized in the section 5.

# E    Reagents Selection

Reagents selection, also known as reagent recommendation, involves the identification and proposal of the most fitting reagents for a specific chemical reaction or process. Compared to other prediction and generation tasks, these selection tasks might be more fitting for LLMs and carry extensive implications. Reagent recommendation can markedly enhance reaction design by pinpointing optimal reagents and conditions for a given reaction, thereby augmenting efficiency and effectiveness in both academic and industrial settings. Drawing from a vast corpus of chemical knowledge, GPT models may be able to generate suggestions, leading to chemical reactions with a greater likelihood of yielding superior results.

In this study, we formulate four reaction component selection task from the Suzuki High-Throughput Experimentation (HTE) dataset. The dataset, created by Perera et al[44](MIT license), evaluates the Suzuki coupling of 5 electrophiles and 7 nucleophiles across a matrix of 11 ligands (with one blank), 7 bases (with one blank), and 4 solvents, resulting in a reaction screening dataset comprising 5,760 data points. The task of reagents selection can be divided into three categories: Reactant selection, Ligand Selection and Solvent selection. For validation, 30 examples were randomly sampled, while 100 examples were used for testing, all taken from the original datasets. Top-1 Accuracy serves as the assessment metric for both reactant and solvent selection, while Top-50% is utilized for ligand selection, as the upper half of the ligands in the list typically provide satisfactory yields in chemical reactions. This task is newly emergent in the field of chemistry, and as such, there are no established baselines yet.

**ICL prompt.** One example of our ICL prompt for reagents selection is shown in Figure 9. Considering the structure of the dataset and the characteristics of the reagents, we provide detailed task description and an answer template to guide GPT models towards the desired output.

**Results.** Our results are presented in Table 12. From the table, it is evident that GPT-4 and GPT-3.5 perform comparatively well in reagent selection tasks. This suggests a promising potential for GPT models in the realm of reagent selection.

Table 12: Accuracy (↑) of LLM in the reagent selection tasks. For Reactant Selection and Solvent selection task, we report the mean (and standard deviation) of the Top-1 Accuracy score and we report the Top-50% accuracy score for the Ligand Selection task. The best LLM is in bold font, and the baseline is underlined.

|  | Reactant Selection | Solvent Selection | Ligand Selection |
|---|---|---|---|
| GPT-4 (zero-shot) | 0.299 ±0.029 | **0.526**±0.012 | **0.534**±0.059 |
| GPT-3.5 (zero-shot) | **0.400**±0.038 | 0.368±0.034 | 0.436 ± 0.020 |
| Davinci-003 (zero-shot) | 0.178 ± 0.034 | 0.463 ± 0.014 | 0.432 ± 0.020 |
| Llama2-13B-chat (zero-shot) | 0.145 ± 0.000 | 0.050 ± 0.010 | 0.284 ± 0.024 |
| GAL-30B (zero-shot) | 0.107 ± 0.020 | 0.104 ± 0.004 | 0.030 ± 0.016 |

# F    Retrosynthesis

Retrosynthesis planning is a crucial task in synthetic organic chemistry that involves identifying efficient synthetic pathways for a target molecule by recursively transforming it into simpler precursor molecules. In contrast to reaction prediction, retrosynthesis planning involves a reverse extrapolation from the target molecule to identify the readily available reactants for its synthesis. In this study, we use the USPTO-50k dataset [53](MIT license), which contains 50,037 chemical reactions. In our

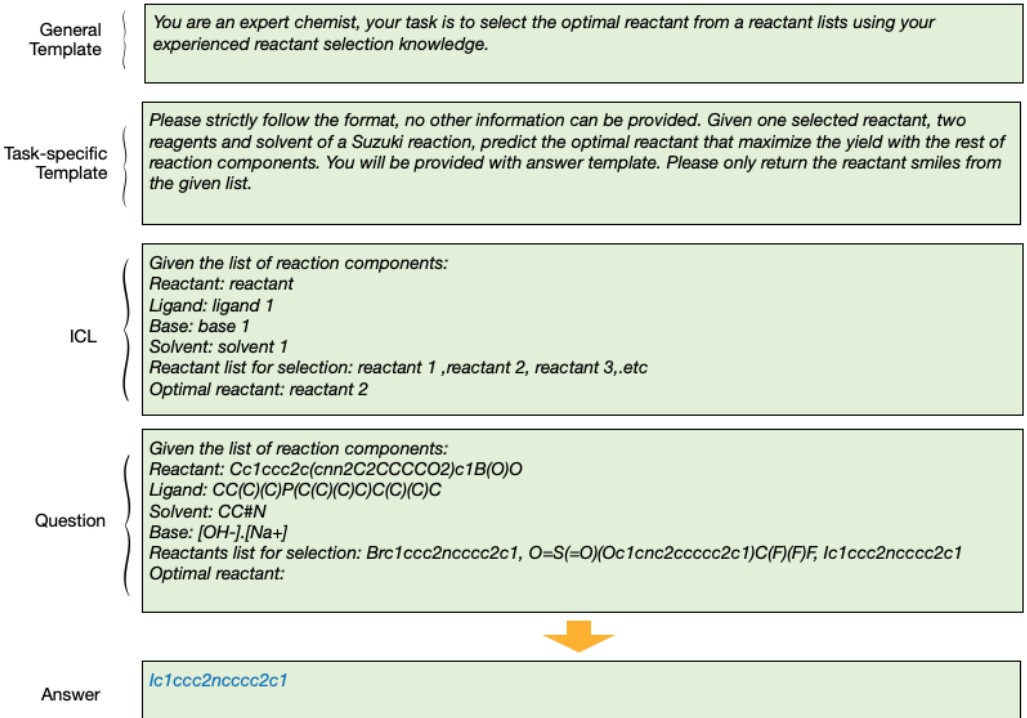

Figure 9: An ICL prompt example for reagents selection

experiment, we use the data splitting as [17] and we the training set which contains 40,029 reactions as the ICL candidates. The metric and baseline are the same as the reaction prediction.

**ICL Prompt.** One example of our ICL prompt for reaction prediction is shown in Figure 10. As discussed in the reaction prediction task, we also add the task-specific template to help GPT models understand the input and restrict the output.

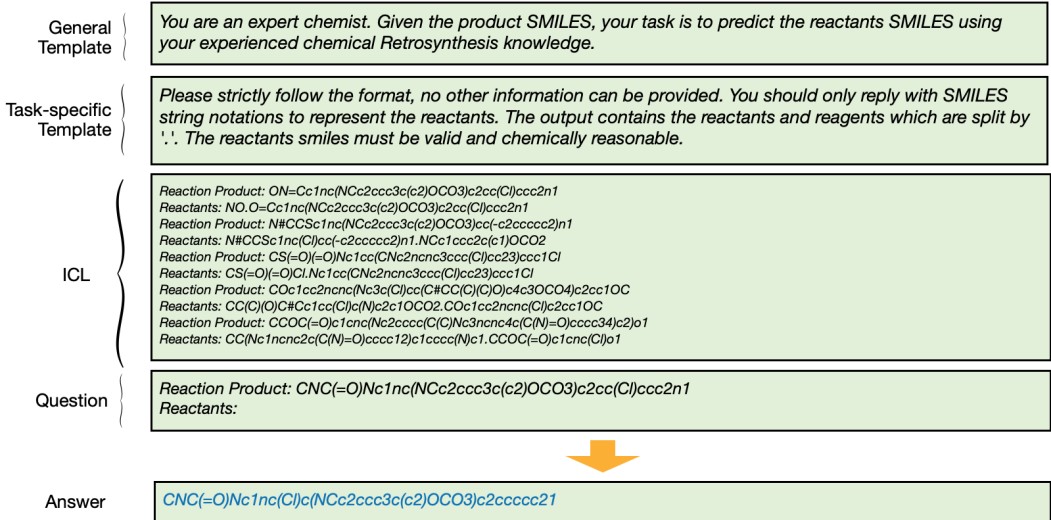

Figure 10: An ICL prompt example for Retrosynthesis

Table 13: The performance of LLMs and baseline in Retrosynthesis task. The best LLM is in bold font, and the baseline is underlined.

| Method | Top-1 Accuracy (↑) | Invalid SMILES (↓) |
|---|---|---|
| Chemformer [26] | 0.536 | 0% |
| GPT-4 (zero-shot) | $0.006 \pm 0.005$ | $20.6\% \pm 4.7\%$ |
| GPT-4 (Scaffold, $k$=20) | $0.096 \pm 0.013$ | $10.4\% \pm 3.4\%$ |
| GPT-4 (Scaffold, $k$=5) | $0.114 \pm 0.013$ | $11.0\% \pm 1.2\%$ |
| GPT-4 (Random, $k$=20) | $0.012 \pm 0.011$ | $18.2\% \pm 4.2\%$ |
| GPT-3.5 (Scaffold, $k$=20) | $0.022 \pm 0.004$ | $6.4\% \pm 1.3\%$ |
| Davinci-003 (Scaffold, $k$=20) | $\mathbf{0.122 \pm 0.013}$ | $6.0\% \pm 1.2\%$ |
| Llama2-13B-chat (Scaffold, k=20) | 0 | $27.2\% \pm 1.5\%$ |
| GAL-30B (Scaffold, k=5) | $0.016 \pm 0.005$ | $\mathbf{5.2\% \pm 1.8\%}$ |

**Results.** The results are reported in Table 13. The performance of GPT models is also inferior than the baseline due to the lack of an in-depth understanding of the SMILES strings that represent reactants and products. Detailed analysis are summarized in the later section 5 Discussion.

## G   Text-Based Molecule Design

Text-Based Molecule Design is a novel task in computational chemistry and drug discovery. It involves generating new molecules with desired molecule descriptions. In our experiment, we employ the ChEBI-20 dataset which consists of 33,010 molecule-description pairs. The dataset is split into 80/10/10% as the training/validation/test set [17](CC BY 4.0). We use the training set which contains 26407 molecule-description pairs as the ICL candidates. For comparison, we use the MolT5-Large [17] as the baseline. MolT5-Large is the initial effort to investigate the translation between molecules and text, including tasks such as text-based molecule design and molecule captioning. It builds upon T5 [46], an encoder-decoder Transformer model, and benefits from pretraining on a large amount of dataset. To comprehensively evaluate the performance, we employ three different types of metrics. The first type of metric is the chemical similarity between the ground-truth molecules and generated molecules, measured by FTS (fingerprint Tanimoto Similarity) [55] in terms of MACCS [49], RDK [35], and Morgan [14]. Secondly, we also use FCD (Fréchet ChemNet Distance) [45] which allows comparing molecules based on the latent information used to predict the activity of molecules [17]. Since the generated molecules are in SMILES string format, we also employ natural language processing metrics including BLEU, Exact Match [17], and Levenshtein distance [40] between the ground-truth molecules and generated molecules SMILES. Finally, to evaluate whether generated molecules are valid, we use RDKIT [35] to check the validity of generated molecules and report the percent of the valid molecules.

**ICL Prompt.** One ICL prompt example for text-based molecule design is shown in Figure 11.

Table 14: The performance of LLMs and baseline in the Text-Based Molecule Design task. The best LLM is in bold font and the baseline is underlined.

| Method | BLEU (↑) | Exact (↑) | Levenshtein (↓) | Validity (↑) | MACCS FTS (↑) | RDK FTS (↑) | Morgan FTS (↑) | FCD (↓) |
|---|---|---|---|---|---|---|---|---|
| MolT5-Large [17] | 0.601 | 0.290 | 41.600 | 0.940 | 0.879 | 0.797 | 0.752 | 5.394 |
| GPT-4 (zero-shot) | $0.490\pm0.017$ | $0.046\pm0.009$ | $47.418\pm1.668$ | $0.758\pm0.015$ | $0.733\pm0.020$ | $0.514\pm0.021$ | $0.432\pm0.014$ | $11.913\pm0.972$ |
| GPT-4 (Scaffold, $k$=10) | $\mathbf{0.816\pm0.004}$ | $\mathbf{0.174\pm0.029}$ | $\mathbf{21.160\pm0.600}$ | $0.888\pm0.023$ | $\mathbf{0.867\pm0.005}$ | $0.738\pm0.010$ | $\mathbf{0.672\pm0.013}$ | $6.224\pm0.449$ |
| GPT-4 (Scaffold, $k$=5) | $0.815\pm0.011$ | $0.164\pm0.018$ | $21.862\pm1.768$ | $0.874\pm0.030$ | $0.865\pm0.015$ | $\mathbf{0.741\pm0.023}$ | $0.670\pm0.028$ | $\mathbf{5.843\pm0.515}$ |
| GPT-4 (Random, $k$=10) | $0.602\pm0.016$ | $0.060\pm0.007$ | $42.390\pm1.008$ | $0.770\pm0.030$ | $0.762\pm0.013$ | $0.548\pm0.017$ | $0.475\pm0.015$ | $10.594\pm0.414$ |
| GPT-3.5 (Scaffold, $k$=10) | $0.479\pm0.156$ | $0.094\pm0.011$ | $82.008\pm40.354$ | $0.854\pm0.059$ | $0.833\pm0.006$ | $0.686\pm0.016$ | $0.585\pm0.013$ | $8.341\pm0.607$ |
| Davinci-003 (Scaffold, $k$=10) | $0.741\pm0.011$ | $0.100\pm0.010$ | $25.648\pm2.186$ | $0.936\pm0.009$ | $0.783\pm0.014$ | $0.648\pm0.004$ | $0.560\pm0.010$ | $8.335\pm0.310$ |
| Llama2-13B-chat (Scaffold, $k$=10) | $0.626\pm0.013$ | $0.020\pm0.000$ | $33.956\pm2.648$ | $0.782\pm0.008$ | $0.679\pm0.015$ | $0.568\pm0.014$ | $0.454\pm0.009$ | $12.387\pm0.437$ |
| GAL-30B (zero-shot) | $0.004\pm0.000$ | $0.000\pm0.000$ | $2738.136\pm166.093$ | $\mathbf{0.956\pm0.011}$ | $0.233\pm0.011$ | $0.109\pm0.006$ | $0.053\pm0.002$ | $35.091\pm0.774$ |

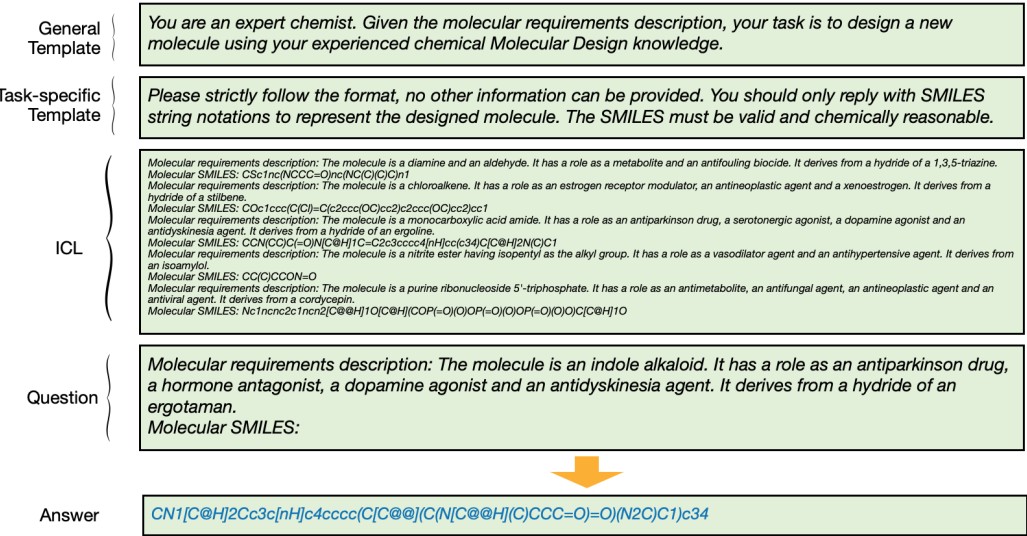

Figure 11: An ICL prompt example for Text-Based Molecule Design

**Results.**    The results are reported in Table 14. We can observe that the best ICL prompting GPT models (GPT-4 and Davinci-003) can achieve competitive performance or even outperform the baseline in some metrics (BLEU, Levenshtein). Although the GPT models significantly underperform the baseline in terms of exact match and Morgan FTS metrics, it's important to note that we only utilize a maximum of 10 examples, which is substantially less than the training set (comprising 26,407 training examples) used for the baseline. These results demonstrate the strong few-shot text-based molecule design ability of GPT models. Last, not being exactly the same as the ground truth doesn't necessarily mean it's incorrect, especially in the context of molecular design. The molecules generated by GPT models may still be useful and can serve as alternatives to the ground truth, given they fulfill the requirements described in the input text and a majority (over 89%) are chemically valid.

**Case studies.**    We select three different types of molecules (organic molecule without rings, organic molecule with ring, and metal atom) as examples, and show the generated molecules in Figure 12. We observe that the structure of molecules generated by the GPT-4 (Scaffold, $k=10$) method is more similar to the ground truth compared to Davinci-003, GPT-4 (zero-shot), and even the baseline. Additionally, for metal atoms design, GPT models outperform the baseline which wrongly generates the SMILES instead of the metal atom. These cases show promising results of the molecule design ability of GPT models. However, evaluating whether the generated molecules are helpful such as molecule novelty in real-world scenarios is still a difficult problem. Thus we conclude that GPT models have excellent potential in molecule design and there are prospects for investigating this ability.

# H    Molecule Captioning

Molecule captioning is an important task in computational chemistry, offering valuable insights and applications in various areas such as drug discovery, materials science, and chemical synthesis. Given a molecule as input, the goal of this task is to generate a textual description that accurately describes the key features, properties, and functional groups of the molecule. We also use the ChEBI-20 dataset(CC BY 4.0) and the training set of it as the ICL candidates as discussed in the Text-Based Molecule Design Section. We use traditional captioning metrics including BLEU, ROUGE, and METEOR for evaluation.

**ICL Prompt.**    One example of our ICL prompt for molecule captioning is shown in Figure 13.

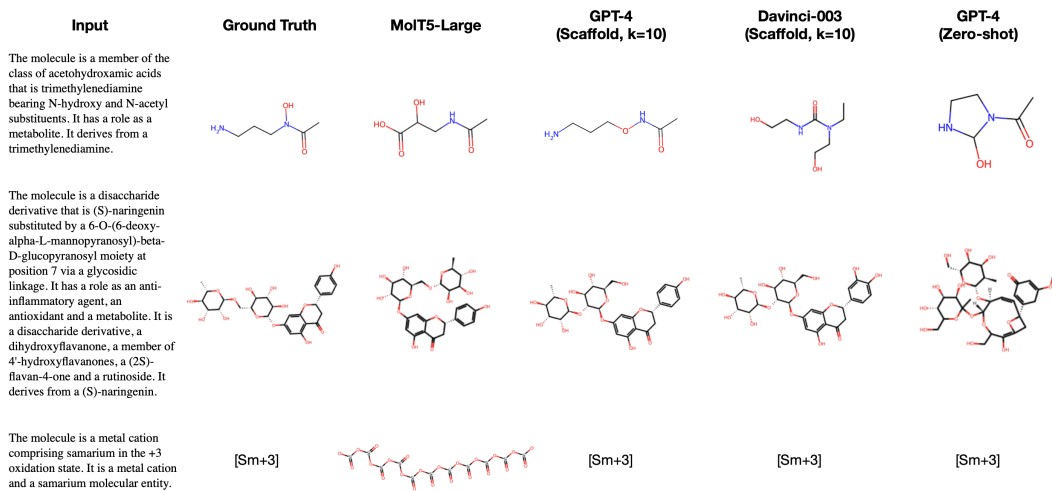

Figure 12: Examples of molecules generated by different models.

**Results.** The results are reported in Table 15. We can observe that the best ICL prompting GPT models (GPT-4 and Davinci-003) can achieve competitive performance or even outperform the baseline in some metrics (BLEU-2 and BLEU-4). This indicates the inspiring capability of the GPT models in the molecule captioning task.

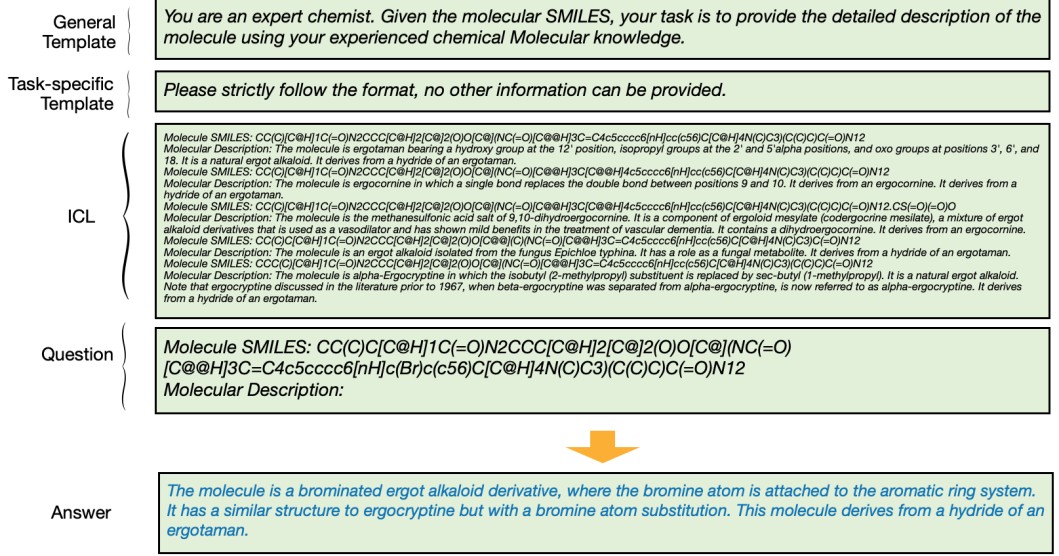

Figure 13: An ICL prompt example for molecule captioning

**Case studies.** Same as case studies in the Text-Based Molecule Design task, we also select three different types of molecules as examples, and the captions are shown in Figure 14. We observe that although the performance of the baseline is close to GPT models, the captions generated by the baseline contain more descriptions that violate the chemical facts. In contrast, the captions generated by GPT-4 models contain only a few inaccurate descriptions, highlighting the excellent explaining ability of GPT models. This highlights the limitations of applying traditional Natural Language Processing (NLP) evaluation metrics to this task. Therefore, it is necessary to create more suitable evaluation metrics for chemistry-related generation tasks.

Table 15: The performance of LLMs and baseline in the molecule captioning task. The best LLM is in bold font and the baseline is underlined.

| Method | BLEU-2 (↑) | BLEU-4 (↑) | ROUGE-1 (↑) | ROUGE-2 (↑) | ROUGE-L (↑) | METEOR (↑) |
|---|---|---|---|---|---|---|
| MolT5-Large [17] | 0.482 | 0.383 | 0.574 | 0.410 | 0.516 | 0.530 |
| GPT-4 (zero-shot) | 0.062±0.001 | 0.013±0.001 | 0.192±0.002 | 0.040±0.002 | 0.125±0.002 | 0.209±0.002 |
| GPT-4 (Scaffold, $k$=10) | 0.464±0.008 | 0.365±0.008 | **0.545±0.003** | **0.362±0.003** | 0.459±0.007 | **0.519±0.005** |
| GPT-4 (Scaffold, $k$=5) | 0.456±0.003 | 0.357±0.004 | 0.540±0.005 | 0.355±0.007 | 0.455±0.005 | 0.505±0.005 |
| GPT-4 (Random, $k$=10) | 0.260±0.007 | 0.140±0.007 | 0.393±0.004 | 0.180±0.006 | 0.309±0.004 | 0.320±0.007 |
| GPT-3.5 (Scaffold, $k$=10) | 0.468±0.010 | 0.368±0.010 | 0.534±0.005 | 0.355±0.007 | 0.457±0.006 | 0.497±0.005 |
| Davinci-003 (Scaffold, $k$=10) | **0.488±0.011** | **0.391±0.012** | 0.532±0.008 | 0.359±0.010 | **0.465±0.008** | 0.478±0.011 |
| Llama2-13B-chat (Scaffold, k=10) | 0.197±0.005 | 0.140±0.004 | 0.331±0.005 | 0.193±0.005 | 0.265±0.005 | 0.372±0.006 |
| GAL-30B (zero-shot) | 0.008±0.000 | 0.002 ± 0.000 | 0.019±0.002 | 0.004±0.000 | 0.015±0.002 | 0.043±0.002 |

Figure 14: Examples captions generated by different models. Descriptions that violate chemical facts are marked in grey.

# I   The comparison of SMILES and SELFIES

Table 16: F1 (↑) score of SMILES and SELFIES of GPT-4 model in molecular property prediction tasks.

| | BBBP | BACE | HIV | Tox21 | ClinTox |
|---|---|---|---|---|---|
| SMILES | $0.587 \pm 0.018$ | $0.666 \pm 0.023$ | $0.797 \pm 0.021$ | $0.563 \pm 0.008$ | $0.736 \pm 0.033$ |
| SELFIES | $0.541 \pm 0.001$ | $0.601 \pm 0.036$ | $0.784 \pm 0.014$ | $0.478 \pm 0.011$ | $0.654 \pm 0.025$ |

Table 17: Performance of SMILES and SELFIES of GPT-4 model in reaction prediction task.

|  | Top-1 Accuracy (↑) | Invalid SMILES/SELFIES (↓) |
|---|---|---|
| SMILES | $0.230 \pm 0.022$ | $7.0\% \pm 1.6\%$ |
| SELFIES | $0.110 \pm 0.007$ | $1.0\% \pm 0.0\%$ |

Table 18: Performance of SMILES and SELFIES of GPT-4 model in molecule design task.

|  | BLEU (↑) | Exact (↑) | Levenshtein (↓) | Validity (↑) | MACCS FTS (↑) | RDK FTS (↑) | Morgan FTS (↑) | FCD (↓) |
|---|---|---|---|---|---|---|---|---|
| SMILES | $0.816 \pm 0.004$ | $0.174 \pm 0.029$ | $21.160 \pm 0.600$ | $0.888 \pm 0.023$ | $0.867 \pm 0.005$ | $0.738 \pm 0.010$ | $0.672 \pm 0.013$ | $6.224 \pm 0.449$ |
| SELFIES | $0.277 \pm 0.009$ | $0.100 \pm 0.016$ | $76.162 \pm 2.229$ | $0.804 \pm 0.022$ | $0.619 \pm 0.010$ | $0.467 \pm 0.018$ | $0.399 \pm 0.017$ | $13.557 \pm 0.224$ |

Table 19: Performance of SMILES and SELFIES of GPT-4 model in molecule captioning task.

|  | BLEU-2 (↑) | BLEU-4 (↑) | ROUGE-1 (↑) | ROUGE-2 (↑) | ROUGE-L (↑) | METEOR (↑) |
|---|---|---|---|---|---|---|
| SMILES | $0.464 \pm 0.008$ | $0.365 \pm 0.008$ | $0.545 \pm 0.003$ | $0.362 \pm 0.003$ | $0.459 \pm 0.007$ | $0.519 \pm 0.005$ |
| SELFIES | $0.459 \pm 0.012$ | $0.367 \pm 0.010$ | $0.530 \pm 0.007$ | $0.360 \pm 0.005$ | $0.456 \pm 0.005$ | $0.490 \pm 0.007$ |

