# OpenReview forum: "What can Large Language Models do in chemistry? A comprehensive benchmark on eight tasks"
_NeurIPS.cc/2023/Track/Datasets_and_Benchmarks — NeurIPS 2023 Datasets and Benchmarks Poster_

### Official Review · Reviewer_K37q · 2023-07-08
**Well-motivated but may need more comprehensive experiments**

**Rating:** 6
**Confidence:** 4
**Correctness:** Yes.
**Clarity:** Yes.

**Strengths:**

1. The work is well-motivated. Benchmarking LLMs on chemical-related tasks can be a valuable contribution to help researchers leverage and improve the power of LLMs in chemical science.
2. The work implements ICL prompt which provides chemical context to improve LLMs' performance.
3. The authors thoroughly discuss benchmarking results and potential lessons that can be learned.

**Additional Feedback:**

N/A

**Documentation:**

Yes.

**Ethics:**

No.

**Limitations:**

The authors have discussed limitations of the work in the manuscript. More potential limitations can be found in the previous section "Opportunities For Improvement".

**Opportunities For Improvement:**

1. The benchmark in this work focuses on property prediction or molecular understanding. However, as the authors mention in Related Works, there is also another category of using LLMs as a planner for chemical experiments/designs. I wonder whether the authors consider adding experiments regarding the other category.
2. The work benchmarks three LLMs, namely GPT-3.5, GPT-4, and Davinci-003. There are also other LLMs like LLaMA and Chinchilla. The authors may add more models to comprehensively benchmark LLMs in chemistry.
3. In ICL, top-k similar molecules are selected. Why not use dissimilar molecules? It seems that dissimilar molecules can give more diverse chemical information to help the tasks.
4. This may be beyond the scope of this work. But current work only considers prompting pretrained LLMs instead of tuning them. Since the models are not trained specifically for chemical-related tasks, prompting alone may not fully release the power of them.
5. As the authors mention, SMILES can lead to invalid molecules. Have the authors considered other string-based representations like SELFIES, which guarantee valid molecules?
6. In name prediction tasks, formula2smiles.may be ill-posed. Since one formula can correspond to different SMILES. It is not surprising to see that all models get 0 accuracy in this task (Table 3).
7. In property prediction, only MoIR is included as a baseline in comparison to LLMs. There are many other domain-specific models that benchmark on MoleculeNet. The authors are suggested to add more baselines to further evaluate the performance of LLMs.

**Relation To Prior Work:**

Yes.

**Summary And Contributions:**

In this work, the authors aim at benchmarking LLMs on chemical-related tasks. In particular, the work includes 3 GPT-styled models, namely GPT-3.5, GPT-4, and Davinci-003. Eight tasks including molecular property prediction, molecular understanding, synthesis prediction, etc are included. The authors also apply in-context learning prompting to improve the performance of LLMs. Experimental results show that LLMs can achieve competitive performance in several tasks yet still struggle in reaction prediction, retrosynthesis, and name prediction.

---

> ### Author Response · Authors · 2023-08-22
> **Response to Reviewer K37q[1/3]**
>
> We thank the reviewer for their feedback and detailed review. Please find our responses to your questions and concerns below. We have added following useful discussion and additional experiments to our revised manuscript.
>
> **Q1: The benchmark in this work focuses on property prediction or molecular understanding. However, as the authors mention in Related Works, there is also another category of using LLMs as a planner for chemical experiments/designs. I wonder whether the authors consider adding experiments regarding the other category.**
>
> > A1: Thank you for highlighting the potential application of LLMs as a planner for chemical experiments/designs. We acknowledge that LLMs hold significant promise in this domain and there are several works in NLP domain focusing on testing the planning ablity of LLMs, for example [1] tested LLMs as an embodied agent, and [2,3], as we mentioned in our paper, tested the capablity of LLMs as an agent to plan chemical synthesis. The goals of our study are distinct as to not duplicate these earlier studies that would have necessitated a considerably different experimental setup and validation that is beyond the scope of our work.
>
> > [1] Song, Chan Hee, et al. "Llm-planner: Few-shot grounded planning for embodied agents with large language models." arXiv preprint arXiv:2212.04088 (2022).
> > [2] Bran, Andres M., et al. "ChemCrow: Augmenting large-language models with chemistry tools." arXiv preprint arXiv:2304.05376 (2023).
> > [3] Boiko, Daniil A., Robert MacKnight, and Gabe Gomes. "Emergent autonomous scientific research capabilities of large language models." arXiv preprint arXiv:2304.05332 (2023).
>
>
> **Q2: The work benchmarks three LLMs, namely GPT-3.5, GPT-4, and Davinci-003. There are also other LLMs like LLaMA and Chinchilla. The authors may add more models to comprehensively benchmark LLMs in chemistry.**
>
> > A2: Thanks for your advice. We have incorporated **Llama2-13b-chat** and **Galactic-30b** into our experimentn(see our updated paper for details). Llama2-13b-chat is trained on a large corpus of publicly available online data, utilizing Reinforcement Learning with Human Feedback (RLHF). On the other hand, Galactica is a general-purpose scientific language model that has been trained on a substantial corpus of scientific text and data. We tested the capability of these models via the same experiment setting and add the results to the original table (highlighted in blue). Chinchilla is a closed source LLM and we do not have access to it. We will progressively add more related LLMs to comprehensively benchmark them in chemistry.
>
>
>
> **Q3: In ICL, top-k similar molecules are selected. Why not use dissimilar molecules? It seems that dissimilar molecules can give more diverse chemical information to help the tasks.**
>
> > A3: Thank you for raising the interesting point about the potential advantages of using dissimilar molecules in the ICL process. The intuition behind using similar molecules was to leverage the shared chemical and structural properties to aid in the prediction tasks. However, we recognize the potential benefits of using dissimilar molecules, as they might introduce a more diverse chemical information that can be beneficial in certain tasks.
> >
> > To explore this, we conducted an additional experiment comparing the performance when selecting top-k similar versus dissimilar molecules. The results are summarized below:
> >
> > Table 1
> > | F1(↑)      | BBBP              | BACE              | HIV               | Tox21             | ClinTox           |
> > | ---------- | ----------------- | ----------------- | ----------------- |:----------------- |:----------------- |
> > | Similar    | $0.587\pm(0.018)$ | $0.666\pm(0.023)$ | $0.797\pm(0.021)$ | $0.563\pm(0.008)$ | $0.736\pm(0.033)$ |
> > | Dissimilar | $0.665\pm(0.026)$ | $0.732\pm(0.039)$ | $0.687\pm(0.014)$ | $0.526\pm(0.036)$ | $0.721\pm(0.029)$ |
> >
> Interestingly, the results show a wide range of effects with some datasets (BBBP, BACE) showing better results while for others (HIV, Tox21), the performance is lower or the same (ClinTox). The origin of these differences is likely to be complex and beyond the scope of the present contribution, but might be a topic of a separate study.

---

> ### Author Response · Authors · 2023-08-22
> **Response to Reviewer K37q[2/3]**
>
> **Q4: This may be beyond the scope of this work. But current work only considers prompting pretrained LLMs instead of tuning them. Since the models are not trained specifically for chemical-related tasks, prompting alone may not fully release the power of them.**
>
> > A4: We agree with the reviewer about tuning to unlock the power of LLMs in chemistry. This is indeed a long-term goal of our work. Our current work mainly focuses on benchmarking LLMs and the main motivation is to build a systematic evaluation datasets and methodology to quantify and evaluate current LLMs in chemistry (similar to BIG-Bench[1]), which is a necessary step before finetuning can be attmepted. While finetuning of some proprietary systems like ChatGPT is not possible, other attempts at finetuning have been made, e.g. [2]. We appreciate for this suggestion and will continually focusing on new LLMs and using the benchmarks in present work to evaluate them.
>
> > [1] Srivastava, et al. Beyond the Imitation Game: Quantifying and extrapolating the capabilities of language models.
> [2] Fang, Yin, et al. Mol-Instructions: A Large-Scale Biomolecular Instruction Dataset for Large Language Models.
>
> **Q5:As the authors mention, SMILES can lead to invalid molecules. Have the authors considered other string-based representations like SELFIES, which guarantee valid molecules?**
>
> > A5: Thank you so much for your advice, we investigated this on four tasks including molecule property prediction, reaction prediction, molecule design and molecule captioning. The experiment results are as follows, indicating **the performance of SELFIES is inferior to the SMILES**. This is in line with expectations as the rules for SELFIES are more complex than those for SMILES.  The detailed discuss can be found in our updated paper, under Session-4.4 - "Are molecule SELFIES representations more suitable for LLMs than SMILES representations?"
>
> >
> >
> > | F1(↑)   | BBBP              | BACE              | HIV               | Tox21             | ClinTox           |
> > | ------- | ----------------- | ----------------- | ----------------- |:----------------- |:----------------- |
> > | SMILES  | $0.587\pm(0.018)$ | $0.666\pm(0.023)$ | $0.797\pm(0.021)$ | $0.563\pm(0.008)$ | $0.736\pm(0.033)$ |
> > | SELFIES | $0.541\pm(0.001)$ | $0.601\pm(0.036)$ | $0.784\pm(0.014)$ | $0.478\pm(0.011)$ | $0.654\pm(0.025)$ |
> >
> > |         | Top-1 Accuracy (↑) | Invalid SMILES/SELFIES (↓) |
> > |---------|--------------------|----------------------------|
> > | SMILES  |    0.230 ± 0.022   |         7.0% ± 1.6%        |
> > | SELFIES |    0.110 ± 0.007   |         1.0% ± 0.0%        |
> >
> >
> > |         | BLEU-2 (↑)      | BLEU-4 (↑)    | ROUGE-1 (↑)   | ROUGE-2 (↑)   | ROUGE-L (↑)   | METEOR (↑)    |
> > |---------|-----------------|---------------|---------------|---------------|---------------|---------------|
> > | SMILES  |  0.464 ± 0.008  | 0.365 ± 0.008 | 0.545 ± 0.003 | 0.362 ± 0.003 | 0.459 ± 0.007 | 0.519 ± 0.005 |
> > | SELFIES |  0.459 ± 0.012  | 0.367 ± 0.010 | 0.530 ± 0.007 | 0.360 ± 0.005 | 0.456 ± 0.005 | 0.490 ± 0.007 |
> >
> > |         | BLEU (↑)        | Exact (↑)     | Levenshtein (↓) | Validity (↑)  | MACCS FTS (↑) | RDK FTS (↑)   | Morgan FTS (↑) | FCD (↓)        |
> > |---------|-----------------|---------------|-----------------|---------------|---------------|---------------|----------------|----------------|
> > | SMILES  |  0.816 ± 0.004  | 0.174 ± 0.029 | 21.160 ± 0.600  | 0.888 ± 0.023 | 0.867 ± 0.005 | 0.738 ± 0.010 | 0.672 ± 0.013  | 6.224 ± 0.449  |
> > | SELFIES |  0.277 ± 0.009  | 0.100 ± 0.016 | 76.162 ± 2.229  | 0.804 ± 0.022 | 0.619 ± 0.010 | 0.467 ± 0.018 | 0.399 ± 0.017  | 13.557 ± 0.224 |

---

> ### Author Response · Authors · 2023-08-22
> **Response to Reviewer K37q[3/3]**
>
> **Q6:In name prediction tasks, formula2smiles.may be ill-posed. Since one formula can correspond to different SMILES. It is not surprising to see that all models get 0 accuracy in this task (Table 3).**
>
> > A6: Thanks so much for this insightful advice. We remove the formula2smiles task and add a new iupac2formula task. One iupac name corresponds to one molecule formula. The experiment results and example for this task are shown below. The accuracy of the best model in this task is also extremely low (0.118), indicating the LLMs still lack basic iupac name understanding ability. We have updated the result to our paper (in Table 4 and Table 5).
>
> | Method                            |  iupac2formula  |
> |-----------------------------------|:---------------:|
> | GPT-4 (zero-shot)                 | 0.092 ± 0.018 |
> | GPT-4 (Scaffold, k=5 )            | 0.118 ± 0.022   |
> | GPT-4 (Scaffold, k=20 )           | 0.084 ± 0.005 |
> | GPT-4 (Random, k=20 )             | 0.076 ± 0.011 |
> | GPT-3.5 (Scaffold, k=20 )         | 0.044 ± 0.009 |
> | Davinci-003 (Scaffold, k=20 )     | 0.018 ± 0.004 |
> | Llama2-13B-chat (Scaffold, k=20 ) |        0        |
> | GAL-30B (Scaffold, k=10 )         |        0        |
>
> | Task          | Input                                                                                        | Ground Truth | Output of GPT-4 (Scaffold, k=20) |
> | ------------- |:-------------------------------------------------------------------------------------------- | ------------ | -------------------------------- |
> | iupac2formula | R)-(1-benzylquinolin-1-ium-4-yl)-(5-ethenyl-1- azabicyclo[2.2.2]octan-2-yl)methanol;chloride | C26H29ClN2O  | C23H27ClN2                       |
>
>
>
> **Q7:In property prediction, only MoIR is included as a baseline in comparison to LLMs. There are many other domain-specific models that benchmark on MoleculeNet. The authors are suggested to add more baselines to further evaluate the performance of LLMs.**
>
> > A7: Thank you for your valuable feedback on the inclusion of additional baselines for our property prediction task. We acknowledge the importance of comprehensive benchmarking to evaluate the performance of LLMs in the context of MoleculeNet.
> >
> > To address this concern, we have now included two widely-used baselines from the MoleculeNet benchmark: Random Forest (RF) and XGBoost. The results from these models are shown below in the table and also updated to our revised manuscript (Table 5). We believe that the addition of these baselines provides a more comprehensive perspective on the performance of LLMs in comparison to domain-specific models.
>
> > | F1(↑)   | BBBP    | BACE    | HIV     | Tox21   | ClinTox |
> > | ------- | ------- |:------- | ------- | ------- |:------- |
> > | RF      | $0.881$ | $0.758$ | $0.518$ | $0.260$ | $0.461$ |
> > | XGBoost | $0.897$ | $0.765$ | $0.551$ | $0.333$ | $0.620$ |
> >

---

> > ### Author Response · Authors · 2023-08-25
> > **Discussion Reminder**
> >
> > Dear Reviewer K37q,
> >
> > I hope this message finds you well. Firstly, we would like to express our gratitude for the valuable feedback you've provided on our submission. We've taken your insights into account and have recently submitted our response.
> >
> > As we're keen to ensure that our revisions address all concerns, we are wondering If there are any further questions or comments you'd like us to address, please do let us know. Your insights are immensely valuable to us, and we want to ensure that all aspects of our work are clear and thorough.
> >
> >
> > We truly appreciate the time and effort you invest in the review process. We're hopeful for any further feedback you might provide, which will undoubtedly enhance the quality and rigor of our work.
> >
> > Thank you again for your continued involvement and guidance. We look forward to your feedback.
> >
> >
> > Warm regards,
> >
> > Authors

---

> > > ### Comment · Reviewer_K37q · 2023-08-26
> > > **Response to rebuttal**
> > >
> > > I thank the authors for answering my questions, which have addressed most of my concerns. The extra experiments have greatly increase the quality and scope of the manuscript. Especially,
> > > 1. added extra LLMs, Llama2-13b-chat and Galactic-30b;
> > > 2. added iupac2formula experiments besides formula2smiles, which shows the limitations of GPT-styled models in chemistry;
> > > 3. added experiments on SELFIES besides SMILES;
> > > 4. added more baselines for property predictions.
> > >
> > > Therefore, I have increased my score to 6.

---

> > > > ### Author Response · Authors · 2023-08-27
> > > > **Thank you**
> > > >
> > > > Thank you so much for reconsidering our submission and adjusting the score, we deeply appreciate your thoughtful review and feedback!

---

### Official Review · Reviewer_eMtr · 2023-07-16
**An interesting work about applying GPT models to chemistry**

**Rating:** 7
**Confidence:** 4
**Clarity:** Yes.

**Strengths:**

This paper makes a first step in solving chemical problems by prompt engineering and powerful large language models like GPT. Experiments are well designed and carefully conducted, and the conclusions drawn from experimental results and analysis are interesting and meaningful. The contribution made by this work will provide a high-quality guide for researchers in chemistry to apply GPT models to their work.

**Additional Feedback:**

No additional feedback.

**Correctness:**

Yes, the evaluation methods and experiment desian are appropriate and performed correctly.

**Documentation:**

Yes. there is sufficient detail to support reproducibility.

**Ethics:**

No ethical concerns.

**Limitations:**

Yes.

**Opportunities For Improvement:**

(1) It is concluded in this work that handling SMILES string representations of molecules is hard for GPT models. Hence, it is interesting to see if another more machine-learning-friendly string representation of molecules, SELFIES [1], are easier to be handled by GPT models. Authors are encourage to evaluate GPT models through replacing SMILES strings by SMILES strings in prompts.
(2) In line 236-243, it is mentioned that GPT models may outperform GNN baselines in molecular property prediction because semantic meaning of labels are included in prompts. Do authors try experiments to verify this hypothesis? Authors may try remove semantic mening, or intensionally give misleading semantic meaning of labels and observe whether the performance of GPT models will decrease.


[1] Self-Referencing Embedded Strings (SELFIES): A 100% robust molecular string representation. Arxiv 1905.13741.

**Relation To Prior Work:**

Yes.

**Summary And Contributions:**

This work tries applying three GPT models to a variety of chemistry tasks. It uses two different prompt engineering to do chemistry tasks by making conversations with GPT models. Experiments show that GPT models can achieve comparable or even better performance than baseline GNN models in some chemistry tasks.

---

> ### Author Response · Authors · 2023-08-22
> **Response to Reviewer eMtr[1/2]**
>
> Thanks a lot for your reviews! Your professional reviews offer us great advice towards writing a more comprehensive and competitive paper! And, we are very encouraged that you found evaluation methods and experiment design are appropriate and performed correctly. Please find our responses to your questions and concerns below. We have added following useful discussion and additional experiments to our revised manuscript.
>
> **Q1: It is concluded in this work that handling SMILES string representations of molecules is hard for GPT models. Hence, it is interesting to see if another more machine-learning-friendly string representation of molecules, SELFIES [1], are easier to be handled by GPT models. Authors are encourage to evaluate GPT models through replacing SMILES strings by SMILES strings in prompts.**
>
> > A1: In response to the reviewers suggestion, we investigated this question for four tasks including molecule property prediction, reaction prediction, molecule design and molecule captioning via the (GPT-4, Scaffold, few-shot prompting) setting. The experiment results are shown in the table below, indicating **the performance of SELFIES is in general inferior to the SMILES**. This is in line with expectations as the rules for SELFIES are more complex than those for SMILES. The only notable exception is that the number of invalid SELFIES is lower than the one for SMILES, again reflecting the inherent properties of SELFIES which are designed to guarantee valid molecules. The detailed discuss can be found in our updated paper, under Session-4.4 - "Are molecule SELFIES representations more suitable for LLMs than SMILES representations?"
> >
> >
> >
> >
> > | F1(↑)   | BBBP              | BACE              | HIV               | Tox21             | ClinTox           |
> > | ------- | ----------------- | ----------------- | ----------------- |:----------------- |:----------------- |
> > | SMILES  | $0.587\pm(0.018)$ | $0.666\pm(0.023)$ | $0.797\pm(0.021)$ | $0.563\pm(0.008)$ | $0.736\pm(0.033)$ |
> > | SELFIES | $0.541\pm(0.001)$ | $0.601\pm(0.036)$ | $0.784\pm(0.014)$ | $0.478\pm(0.011)$ | $0.654\pm(0.025)$ |
> >
> > |         | Top-1 Accuracy (↑) | Invalid SMILES/SELFIES (↓) |
> > |---------|--------------------|----------------------------|
> > | SMILES  |    0.230 ± 0.022   |         7.0% ± 1.6%        |
> > | SELFIES |    0.110 ± 0.007   |         1.0% ± 0.0%        |
> >
> >
> > |         | BLEU-2 (↑)      | BLEU-4 (↑)    | ROUGE-1 (↑)   | ROUGE-2 (↑)   | ROUGE-L (↑)   | METEOR (↑)    |
> > |---------|-----------------|---------------|---------------|---------------|---------------|---------------|
> > | SMILES  |  0.464 ± 0.008  | 0.365 ± 0.008 | 0.545 ± 0.003 | 0.362 ± 0.003 | 0.459 ± 0.007 | 0.519 ± 0.005 |
> > | SELFIES |  0.459 ± 0.012  | 0.367 ± 0.010 | 0.530 ± 0.007 | 0.360 ± 0.005 | 0.456 ± 0.005 | 0.490 ± 0.007 |
> >
> > |         | BLEU (↑)        | Exact (↑)     | Levenshtein (↓) | Validity (↑)  | MACCS FTS (↑) | RDK FTS (↑)   | Morgan FTS (↑) | FCD (↓)        |
> > |---------|-----------------|---------------|-----------------|---------------|---------------|---------------|----------------|----------------|
> > | SMILES  |  0.816 ± 0.004  | 0.174 ± 0.029 | 21.160 ± 0.600  | 0.888 ± 0.023 | 0.867 ± 0.005 | 0.738 ± 0.010 | 0.672 ± 0.013  | 6.224 ± 0.449  |
> > | SELFIES |  0.277 ± 0.009  | 0.100 ± 0.016 | 76.162 ± 2.229  | 0.804 ± 0.022 | 0.619 ± 0.010 | 0.467 ± 0.018 | 0.399 ± 0.017  | 13.557 ± 0.224 |
> >

---

> > ### Author Response · Authors · 2023-08-22
> > **Response to Reviewer eMtr[2/2]**
> >
> > **Q2: In line 236-243, it is mentioned that GPT models may outperform GNN baselines in molecular property prediction because semantic meaning of labels are included in prompts. Do authors try experiments to verify this hypothesis? Authors may try remove semantic mening, or intensionally give misleading semantic meaning of labels and observe whether the performance of GPT models will decrease.**
> >
> > > A2: Thank you for your suggestion! We have incorporated experiments(under Appendix B - "Label interpretation") that involve removing label context from the in-context learning prompt in two datasets—HIV and ClinTox. In our testing, we evaluated the GPT-4 model under both few-shot (using scaffold sampling) and zero-shot settings. Our results indicate that when the label information—such as "Activity Test Result" for the HIV dataset, and "FDA-approved" for the ClinTox dataset—is removed from the prompts, the prediction performance of the GPT-4 model noticeably declines as demonstrated in Table 1 and Table 2. This further supports our hypothesis. We have incorporated this result into the revised section titled "Label Interpretation" found in Appendix B (Molecule Property Prediction).
> > >
> > >
> > >
> > >
> > > Table 1:
> > > |          Accuracy(↑)         | HIV               | ClinTox           |
> > > | ---------------------------- |:----------------- | ----------------- |
> > > | GPT-4(zero-shot)             | $0.986\pm(0.07)$  | $0.736\pm(0.027)$ |
> > > | GPT-4(unlabelled, zero-shot) | $0.628\pm(0.016)$ | $0.602\pm(0.039)$ |
> > > | GPT-4(few-shot)              | $0.836\pm(0.020)$ | $0.856\pm(0.014)$ |
> > > | GPT-4(unlabelled, few-shot)  | $0.541\pm(0.032)$ | $0.630\pm(0.014)$ |
> >
> > >Table 2:
> > > | F1(↑)                        | HIV               | ClinTox           |
> > > |:---------------------------- |:----------------- | ----------------- |
> > > | GPT-4(zero-shot)             | $0.977\pm(0.013)$ | $0.489\pm(0.018)$ |
> > > | GPT-4(unlabelled, zero-shot) | $0.554\pm(0.017)$ | $0.438\pm(0.045)$ |
> > > | GPT-4(few-shot)              | $0.797\pm(0.021)$ | $0.563\pm(0.008)$ |
> > > | GPT-4(unlabelled, few-shot)  | $0.493\pm(0.030)$ | $0.478\pm(0.035)$ |

---

> > > ### Comment · Reviewer_eMtr · 2023-08-23
> > > **Response**
> > >
> > > I appreciate authors' hard work in conducting new experiments to address my concerns. Now all my concerns have been addressed. I will keep my acceptance decision.

---

> > > > ### Author Response · Authors · 2023-08-23
> > > > **Thank you**
> > > >
> > > > Thank you very much for taking the time to thoroughly review our manuscript and for recognizing the efforts we put into addressing your concerns. We are grateful for your constructive feedback!

---

### Official Review · Reviewer_5TAm · 2023-07-17
**Very interesting topic to look into, but could have more novelty in the appraoch**

**Rating:** 7
**Confidence:** 3
**Correctness:** Yes
**Clarity:** Yes

**Strengths:**

Evaluating large language models on chemistry tasks is a very interesting topic in the area of combining AI, LLM and science. The authors have evaluated 3 LLMs on 8 chemistry related tasks covering different category of tasks. The results are well presented.

**Additional Feedback:**

No additional feedback.

**Documentation:**

Yes

**Limitations:**

I think there is a lack of significance of novelty with the current work. It is a very interesting topic to discuss to evaluate and get some conclusion. But the whole paper reads more like a technical report. I would expect more original ideas to be presented.

**Opportunities For Improvement:**

1, it would be good to include more LLMs, especially if there are LLMs models that are fine-tuned on chemistry data.
2, it would be good to see more results with advanced prompting techniques like CoT, decomposed prompting, comparing with ICT.
would be good to have some discussion on the following two questions:
3, if there is such parameter, what is the temperature set for the LLMs tested. how would the temperature influence the number of tests run (currently it is 5 tests)?
4, how much pre-training data is related to chemistry for those LLMs? Is there chemistry text-book available in the pre-training dataset? If there is not enough of such chemistry dataset, would there be a limitation for the model to understand and generate chemistry related contents?

**Relation To Prior Work:**

Yes

**Summary And Contributions:**

The authors evaluated the capabilities of LLMs in chemistry. They evaluated 3 LLM models on 8 different tasks covering 3 categories. They have found that GPT-4 outperforms other models on chemistry tasks and LLMs can achieve more satisfactory results with competitive tasks like classification or ranking. Furthermore, they proposed several strategies of prompting.

---

> ### Author Response · Authors · 2023-08-22
> **Response to Reviewer 5TAm[1/3]**
>
> We thank the reviewer for their positive feedback and detailed review. Please find our responses to your questions and concerns below. We added discussion and additional experiments to the updated draft.
>
> **Q1: it would be good to include more LLMs, especially if there are LLMs models that are fine-tuned on chemistry data.**
>
> > Answer: Thank you for your suggestion. We have expanded the scope of the work to incorporate **Llama2-13b-chat** and **Galactic-30b** into our experiment (the experiments result  have been updated to the revised manuscript). Llama2-13b-chat is trained on a large corpus of publicly available online data, utilizing Reinforcement Learning with Human Feedback (RLHF). In contrast, Galactica is a general-purpose scientific language model that has been trained on a substantial corpus of scientific (though not necessarily chemistry-centric) text and data. We tested the capability of these models using the few-shot (scaffold sampling) in-context learning scheme and the result for each task has been added to our revised paper. **Overall, these open-sourced models underperform GPT models.** Our analysis suggests two primary reasons for this observation:
> > >     1). Insufficient Chemical Expertise: The open-sourced models haven't been rigorously trained on expansive chemical datasets. Consequently, they lack a comprehensive understanding of chemical theories, reactions, properties, and specific chemical notations like SMILES, Formula, IUPAC names, and so forth.
>
> > >     2). Inadequate Training Alignment: Unlike GPT models, which have benefited from extensive human feedback during their training phase, these open-sourced counterparts might not have undergone similar interaction. This leads to their diminished capacity to accurately process and address chemical-centric inquiries, thus undermining their efficacy in chemistry-related tasks.
>
>
> **Q2: it would be good to see more results with advanced prompting techniques like CoT, decomposed prompting, comparing with ICT. would be good to have some discussion on the following two questions:**
>
> > A2: Thanks for this insightful advice. Designing advanced prompting techniques for **each task** to evaluate the performance of **each LLM** would be very challenging and we regard it as our long-term goal. Since advanced prompting highly depends on the specific LLMs to use on specific tasks, many previous famous LLM benchmark work such as BIG-Bench [1] and Chain-of-thought hub [2] also used promts based on the best of experts' knowledge, and leave the design of advanced prompting for future study.  Moreover, some advanced prompting ways may be helpful for one LLM but hurt the performance of others. Thus, this would cause unfair evaluation in the benchmark. Due to these challenges, our prompting design considers the key factors: fair and general for most LLMs, human easy-written and understanding, producing stable and best possible results.
>
> > Based on this, in our previous experiments, we have tried different kinds of prompts on our validation set. Our prompting is a **trade-off between "general for most LLMs" and "producing stable and best possible results"**. According to the categorization of LLM prompts  in  [3], our prompting is in the most detailed level which consists of role-play description, task description, guidance of the few-shot examples, etc. We thus believe that  our prompting can achieve best possible results, and also is fair and general for most LLMs.
>
> > Overall, we appreciate for this advice and we will continually update our prompting to build a better and fair LLM benchmark for chemistry.
>
> > [1] A Srivastava, et al. Beyond the Imitation Game: Quantifying and extrapolating the capabilities of language models.
> [2] Yao Fu, et al. Chain-of-Thought Hub: A Continuous Effort to Measure Large Language Models' Reasoning Performance.
> [3] Shubhra Kanti Karmaker Santu et al. TELeR: A General Taxonomy of LLM Prompts for Benchmarking Complex Tasks.

---

> > ### Author Response · Authors · 2023-08-22
> > **Response to Reviewer 5TAm[2/3]**
> >
> > **Q3: if there is such parameter, what is the temperature set for the LLMs tested. how would the temperature influence the number of tests run (currently it is 5 tests)?**
> >
> > > A3: We determined the optimal temperature parameter through evaluation on the validation set. For this process, we randomly sampled 30 data samples from the datasets and conducted in-context learning experiments to assess performance under various temperature settings for token-cost saving and API query time efficiency. To close the gap between validation set and test set, we conducted selective temperature testing directly on the test set (GPT-4, scaffold sampling with k=8) and presented our results for two molecular property prediction datasets, BBBP and BACE. The result are shown in Table 1 and Table 2 and our observations suggest that changes in the temperature parameter result in minimal impact on the test outcomes, with variations of less than 0.05 observed in both accuracy and F1 score. The temperatures we used were  t = [0.2, 0.4, 0.6, 0.8, 1]. The impact of temperature has been updated in Session-4.5, "The impact of temperature parameters of LLMs".
> >
> > > Table 1
> > >
> > > | F1(↑)          | BBBP              | BACE              |
> > > |----------------|-------------------|-------------------|
> > > | GPT-4(t=0.2)   | 0.667±0.029       | 0.741±0.019       |
> > > | GPT-4(t=0.4)   | 0.712±0.014       | 0.728±0.024       |
> > > | GPT-4(t=0.6)   | 0.683±0.016       | 0.736±0.020       |
> > > | GPT-4(t=0.8)   | 0.686±0.030       | 0.744±0.025       |
> > > | GPT-4(t=1.0)   | 0.684±0.023       | 0.756±0.025       |
> >
> > > Table 2
> > > | Accuracy(↑)   | BBBP              | BACE              |
> > > |---------------|-------------------|-------------------|
> > > | GPT-4(t=0.2)  | 0.650±0.028       | 0.743±0.019       |
> > > | GPT-4(t=0.4)  | 0.691±0.017       | 0.729±0.024       |
> > > | GPT-4(t=0.6)  | 0.659±0.016       | 0.736±0.019       |
> > > | GPT-4(t=0.8)  | 0.661±0.032       | 0.745±0.025       |
> > > | GPT-4(t=1.0)  | 0.660±0.021       | 0.757±0.025       |

---

> > > ### Author Response · Authors · 2023-08-22
> > > **Response to Reviewer 5TAm[3/3]**
> > >
> > > **Q4: how much pre-training data is related to chemistry for those LLMs? Is there chemistry text-book available in the pre-training dataset? If there is not enough of such chemistry dataset, would there be a limitation for the model to understand and generate chemistry related contents?**
> > > >
> > > > A4: For the LLMs released by OpenAI (GPT-4, GPT-3.5, etc), the pre-training dataset has not been published. For LLama-2, the concrete pre-training dataset is also not published. Based on Llama-1 reports [1], the pre-training data is shown below. While the provided information lacks detail, we hypothesize that it might encompass a modest portion of text related to chemistry, yet it likely contains an insignificant amount of content from chemistry textbooks.
> > > >
> > > > |    Dataset    | Sampling prop. | Disk size |
> > > > |:-------------:|:--------------:|:---------:|
> > > > |  CommonCrawl  |      167.0     |   3.3 TB  |
> > > > |       C4      |      15.0      |   783 GB  |
> > > > |     Github    |      4.5%      |   328 GB  |
> > > > |   Wikipedia   |      4.5%      |   83 GB   |
> > > > |     Books     |      4.5%      |   85 GB   |
> > > > |     ArXiv     |      2.5%      |   92 GB   |
> > > > | StackExchange |      2.0%      |   78 GB   |
> > >
> > > > For Galactica [2], the pre-training dataset contains more corpus related to chemistry compared to Llama-1 [1]. The following is the part of knowledge bases used in Galactica. More details can be seen in the reports of Galactica. The pre-trianing dataset of this model contains information on a large number of molecules from PubChem, a large public database of compounds and selected properties.
> > > >
> > > > | Data source      |   Documents |       Tokens |
> > > > |------------------|------------:|-------------:|
> > > > | PubChem Compound | 1.7 million |    1 billion |
> > > > | UniProt          |     551,837 |  0.6 billion |
> > > > | RefSeq Genome    |          69 |  0.1 billion |
> > > > | OEIS             |     350,833 | 0.07 billion |
> > > > | Ribosome         |       9,950 | 0.05 billion |
> > > > | LIPID MAPS       |      45,273 | 0.03 billion |
> > > > | Reactome         |         156 | 0.01 billion |
> > > > | NASA Exoplanet   |       5,021 | 0.01 billion |
> > > > | Total            |   2 million |    2 billion |
> > >
> > > > Based on the subpar performance of LLama, we can see that without sufficient chemistry dataset, there will definitely be a limitation for the model to understand and generate chemistry related contents. However, even with enough chemistry-specific datasets, LLMs might still find challenges in certain tasks, as evidenced by Galactica. Our observations suggest that Galactica occasionally struggles with understanding human instructions, likely due to its lack of instructional tuning and alignment with human patterns of communication. Thus, while chemistry-centric pre-training datasets are crucial, we assum that instructional tuning and aligning models more closely with human feedback are equally important for the success of LLMs.
> > >
> > > > [1] Hugo Touvron, et al. LLaMA: Open and Efficient Foundation Language Models.
> > > > [2] Ross Taylor, et al. GALACTICA: A Large Language Model for Science.
> > >
> > >
> > > **Limitations:
> > > I think there is a lack of significance of novelty with the current work. It is a very interesting topic to discuss to evaluate and get some conclusion. But the whole paper reads more like a technical report. I would expect more original ideas to be presented.**
> > > >
> > > > Thank you for raising the concern. Prior to our study, similar efforts [1] primarily documented isolated experiments on a few specific examples using a single large language model. In contrast, our principal contributions, outlined in lines 59-75, center on creating a holistic and systematic benchmark for assessing LLM capabilities. Our efforts go beyond mere observations, focusing on defining the precise roles of LLMs in chemistry, the task selection criteria, and designing a robust and comprehensive experimental methodology. Our goal isn't just to present technical findings but to establish an evaluation framework that paves the way for the effective use of LLMs in the domain of chemistry research.
> > > >
> > > > [1] Cayque Monteiro Castro Nascimento et al. Do Large Language Models Understand Chemistry? A Conversation with ChatGPT.
> > > >

---

> > > > ### Comment · Reviewer_5TAm · 2023-08-26
> > > > **Thanks, I'm in general happy with the revised version of the paper**
> > > >
> > > > Thanks for the response to my review comments, the author has addressed all of my comments and I'm happy with most of them. It is interesting to see more LLMs being added and the impact of temperature on results. I'm looking forward to more prompting techniques in the future. I'd like to increase my score to 7.

---

> > > > > ### Author Response · Authors · 2023-08-27
> > > > > **Thank you**
> > > > >
> > > > > Thank you for your positive feedback and for adjusting the score ! We truly value your insights and look forward to exploring more prompting techniques in future work.

---

> > > > > ### Author Response · Authors · 2023-08-28
> > > > > **Thanks and kind reminder of the score adjusting**
> > > > >
> > > > > Dear Reviewer 5TAm,
> > > > >
> > > > > Thanks so much for your insightful feedback. We truly appreciate the score increase mentioned in your response. Just kind remind that the score hasn't been modified yet and you can modify the previous score directly.
> > > > >
> > > > > Thank you again for your continued involvement and guidance.
> > > > >
> > > > >
> > > > > Warm regards,
> > > > >
> > > > > Authors

---

> > > > > > ### Comment · Reviewer_5TAm · 2023-08-29
> > > > > > **Score increased to 7**
> > > > > >
> > > > > > Yeah sorry about that, now the score has been raised to 7.

---

> > > > > > > ### Author Response · Authors · 2023-08-29
> > > > > > > **Thank you**
> > > > > > >
> > > > > > > Thanks so much for your time!

---

### Official Review · Reviewer_FiPp · 2023-07-21
**An Accessible and Impactful Benchmark**

**Rating:** 8
**Confidence:** 4

**Strengths:**

&nbsp;

The benchmark covers a broad range of tasks and the authors have spent a great deal of attention in making the interface accessible to users e.g. the Jupyter notebooks.

&nbsp;

**Additional Feedback:**

&nbsp;

Discussed above.

&nbsp;

**Clarity:**

&nbsp;


In general the paper is well-written and easy to follow. I have the following additions:

&nbsp;

1. In the title, what is the purpose of including "Indeed". I would suggest removing it.

2. Additionally in the title, it would be more correct to say LLMs in place of GPT models seeing as the benchmark is general, although I understand the authors may wish to retain GPT somewhere in the title to appear higher on search engine results!

3. It would be great if the references appeared in numbered order.

4. Line 356, typo, 'understanding of molecular SMILES strings'.

5. There is missing capitalisation in some of the references e.g. got in place of GPT.

6. Line 166, the number of bits for the fingerprints could also be given. Were the fingerprints Morgan or ECFP? How were they computed, using RDKit?

7. Line 191, typo, "MolR" in place of "MoIR"?

8. Line 671, typo, capitalisation of 'Discussion'.

9. Line 217, typo, molecular formula.

10. In Table 3 of the appendix, what are the unit for the errors?

11. Line 619, typo, "consist of" or "comprise".

&nbsp;

**Correctness:**

&nbsp;

Discussed above.

&nbsp;

**Documentation:**

&nbsp;

In general the codebase is well-documented with particular attention given to usability. I would suggest the following improvement:

&nbsp;

1. In the repository it would be great if the READMEs of the data subfolders could be updated to refer back to the instructions for loading the larger datasets such as USPTO. This may aid the user in navigation.

&nbsp;

**Ethics:**

&nbsp;

No ethical concerns identified.

&nbsp;

**Limitations:**

&nbsp;

Discussed above.

&nbsp;

**Opportunities For Improvement:**

&nbsp;

__**MINOR POINTS**__

&nbsp;

1. Line 51, the authors claim that their benchmark covers a wide spectrum of the chemistry domain (which is a tall order for any benchmark!). While the benchmark is valuable one of the limitations is that there are many areas not covered e.g. spectroscopy and perhaps this could be mentioned as a limitation.

2. In Section 2 on Related Work, it may also be worth mentioning that ChatGPT has struggled on graduate level mathematics problems [1] which also underscores the complexity of achieving task-specific accuracy with LLMs and corroborates the finding of the authors that GPT-4 outperforms GPT-3 on a broad range of tasks. There are also interesting parallels between the difficulty that LLMs have in parsing complex syntax in mathematics and chemistry. For example, LLMs struggle somewhat with the Polish notation for integrals in [1] and they are also found to struggle with string-based representations of molecules such as SMILES in the current work.

3. Line 120, 'taking the suggestions from chemists'. How were these suggestions obtained?

4. Line 179, it would be great if the standard error was reported in place of the variance.

5. Line 200, who were the chemists used to evaluate Figure 13?

6. It may be worth considering adding the photoswitch benchmark dataset from [2] to the property prediction task seeing as it is evaluated in a related work [3] that evaluates GPT-3 on the task of property prediction.

&nbsp;

__**REFERENCES**__

&nbsp;

[1] Frieder et al. 2023. [Mathematical Capabilities of ChatGPT](https://arxiv.org/abs/2301.13867). arXiv preprint arXiv:2301.13867.

[2] Griffiths et al. 2022. [Data-Driven Discovery of Molecular Photoswitches with Multioutput Gaussian Processes](https://pubs.rsc.org/en/content/articlehtml/2022/sc/d2sc04306h). Chemical Science, 13(45), pp.13541-13551.

[3] Jablonka et al. 2023. [Is GPT-3 all you need for Low-Data Discovery in Chemistry?](https://chemrxiv.org/engage/chemrxiv/article-details/63eb5a669da0bc6b33e97a35?trk=public_post_comment-text), ChemRxiv.

&nbsp;

**Relation To Prior Work:**

&nbsp;

Prior work on LLMs in chemistry is adequately discussed. One more recent study concurrent to the submission is [4] which might be included in the camera-ready version of the paper. Additionally when referencing the USPTO dataset, the patent-mining work of Daniel Lowe [5] should be cited additionally as a key source.

&nbsp;

__**REFERENCES**__

&nbsp;

[4] Jablonka et al. [14 Examples of How LLMs Can Transform Materials Science and Chemistry: A Reflection on a Large Language Model Hackathon](https://arxiv.org/abs/2306.06283). arXiv preprint arXiv:2306.06283 (2023).

[5] Lowe, D.M., 2012. [Extraction of Chemical Structures and Reactions from the Literature](https://www.repository.cam.ac.uk/items/dbb4f258-8f3c-4b59-9b5c-62fac7ca8c28) (Doctoral dissertation, University of Cambridge).

&nbsp;

**Summary And Contributions:**

&nbsp;

The authors present an LLM benchmark on 8 chemistry tasks of practical importance. The benchmark is highly accessible with a detailed GitHub repository and there are a number of interesting scientific findings such as the benefits afforded by In-Context Learning (ICL). I believe the benchmark to be a solid contribution and as such, recommend acceptance with points the authors may wish to consider detailed below. I am willing to increase my score if the issues are addressed.

&nbsp;

---

> ### Author Response · Authors · 2023-08-22
> **Response to Reviewer FiPp[1/2]**
>
> We thank the reviewer for their positive feedback and detailed review. Please find our responses to your questions and concerns below. We included the useful discussion and additional experiments in the updated version of our paper.
>
> **Q1: Line 51, the authors claim that their benchmark covers a wide spectrum of the chemistry domain (which is a tall order for any benchmark!). While the benchmark is valuable one of the limitations is that there are many areas not covered e.g. spectroscopy and perhaps this could be mentioned as a limitation**
>
> > A1: We appreciate the critical comment. Our use of the term "wide spectrum" in line 51 is indeed ambitious. we acknowledge that our benchmark, while covering many aspects of chemistry, does not comprehensively cover all areas of the chemistry domain. For instance, as you rightly pointed out, our benchmark does not delve into important areas such as spectroscopy. The field of chemistry is vast and complex, and any benchmark, including ours, necessarily captures only a subset of this domain. Nevertheless, we would like to point out that to the best of our knowledge, our benchmark study is the most comprehensive one presented so far, which is acknowledged by several reviewers. In response to this comment, we have specified the scope of the study more precisely in section 2 (Related Work).
>
> **Q2: In Section 2 on Related Work, it may also be worth mentioning that ChatGPT has struggled on graduate level mathematics problems [1] which also underscores the complexity of achieving task-specific accuracy with LLMs and corroborates the finding of the authors that GPT-4 outperforms GPT-3 on a broad range of tasks. There are also interesting parallels between the difficulty that LLMs have in parsing complex syntax in mathematics and chemistry. For example, LLMs struggle somewhat with the Polish notation for integrals in [1] and they are also found to struggle with string-based representations of molecules such as SMILES in the current work.**
>
> > A2: Thank you for your feedback. We have updated the "Related Work" section and added the relevant references as suggested.
>
> **Q3: Line 120, 'taking the suggestions from chemists'. How were these suggestions obtained?**
>
> > A3: Our work is funded by the NSF Center for Computer Assisted Synthesis (C-CAS, https://ccas.nd.edu/), which brings together over 50 chemists and computer scientists (specialized in AI). In C-CAS, we work closely with  experienced chemists (e.g., those listed among the authors) to identify tasks that can be empowered by large language models. These chemists are largely invovled in the selection of datasets, design of prompts, evaluation of the results and finally paper writing.
>
> **Q4: Line 179, it would be great if the standard error was reported in place of the variance**
>
> > A4: Thanks for pointing out this mistake. The use of the term "variance" in L179 is a typo and we use the "standard deviation" previously. We have fixed it in the paper.
>
> **Q5: Line 200, who were the chemists used to evaluate Figure 13?**
>
> > A5: Apologies for any confusion caused. In Figure 13, we engaged several senior Ph.D. students from Professor Olaf Wiest's group in the Department of Chemistry&Biochemistry to evaluate the results. These evaluations, led by coauthor Bozhao Nan, were conducted independently by each student. After that, a group discussion was held to reconcile their assessments. The results presented in Figure 13 reflect the consensus reached by these students after this discussion.

---

> ### Author Response · Authors · 2023-08-22
> **Response to Reviewer FiPp[2/2]**
>
> **Q6: It may be worth considering adding the photoswitch benchmark dataset from [2] to the property prediction task seeing as it is evaluated in a related work [3] that evaluates GPT-3 on the task of property prediction.**
>
>
> > A6: Thank you for your advice. We added an experiment using the Photoswitch [1] dataset. Following [2], we converted the task into a classification problem by binning the E isomer pi-pi wavelength into five classes. We then sampled 20 data points equally from each class and conducted few-shot (scaffold, k=8) in-context learning experiments using the Davinci-003, GPT-3.5, GPT-4 models. The results of these experiments are presented in Table 1. This was done using the following prompt:
> >
> >* "You are an expert chemist, your task is to predict the property of molecule using your experienced chemical property prediction knowledge.Please strictly adhere to the format, and no other information will be provided. Using the SMILES string representation of a molecule, your task is to determine if the molecule falls into a specific class based on its continuous value. Based on previous binned data, you are to categorize the molecules into one of five classes. The classes represent distinct intervals of molecular properties, specifically the pi-pi* and transition wavelength of the photoswitch.You will be provided with several examples. For each given molecule, determine its class ranging from 1 to 5. Please provide the answer in numeric form corresponding to the class."
>
> >Table 1:result of in-context learning on Photoswitch dataset
> > | Photoswitch               | Accuracy(↑)       | F1(↑)                 |
> > |:------------------------- | ----------------- |:----------------- |
> > | GPT-4(Scaffold,k=8)       | $0.410\pm(0.154)$ | $0.117\pm(0.013)$ |
> > | GPT-3.5(Scaffold,k=8)     | $0.066\pm(0.014)$ | $0.079\pm(0.018)$ |
> > | Davinci-003(Scaffold,k=8) | $0.188\pm(0.009)$ | $0.169\pm(0.009)$ |
>
> > The results in Table 1 reveal that GPT models struggle to accurately predict the wavelength range class to which a photoswitch molecule belongs. This suggests that the models, despite their sophisticated architectures, may lack essential chemical knowledge or contextual understanding required for this specific task. It's also plausible that the in-context learning approach, while useful for some property prediction datasets, isn't optimal for the intricacies of the Photoswitch dataset. The discrepancies in performance across models underscore the challenges faced in adapting language models to specialized scientific domains and emphasize the need for more domain-specific fine-tuning[2] or training strategies. We are grateful for the insights that allowed us to discern the limitations of GPT models on this new dataset.
>
> > [1] Griffiths et al. 2022. Data-Driven Discovery of Molecular Photoswitches with Multioutput Gaussian Processes. Chemical Science, 13(45), pp.13541-13551.
>
> > [2] Jablonka et al. 2023. Is GPT-3 all you need for Low-Data Discovery in Chemistry?, ChemRxiv.
>
> **Clarity:**
>
> > Thank you for your constructive feedback, which we appreciate greatly. We acknowledge your suggestions and corrections to improve the clarity of our paper and have implemented changes accordingly.
>
> **Relation To Prior Work:**
>
> **Prior work on LLMs in chemistry is adequately discussed. One more recent study concurrent to the submission is [4] which might be included in the camera-ready version of the paper. Additionally when referencing the USPTO dataset, the patent-mining work of Daniel Lowe [5] should be cited additionally as a key source.**
>
> > Thank you for the advice, we have added those citations to our revised paper as suggested.
>
> **Documentation**
>
> > Thank you for the advice, we have updated the README of the dataset subfolder to help provide link about datasets we used including USPTO-Mixed, USPTO-50k, etc.

---

> > ### Comment · Reviewer_FiPp · 2023-08-22
> > **Many Thanks for the Additional Experiments and for Addressing all Concerns**
> >
> > &nbsp;
> >
> > Many thanks to the authors for their comprehensive response! I am increasing my score to 8.

---

> > > ### Author Response · Authors · 2023-08-23
> > > **Thank you**
> > >
> > > Thank you a lot for your help in strengthening the paper and for raising the score!

---

### Official Review · Reviewer_nBsG · 2023-07-31
**Good paper in understanding LLM's capabilities in Chemistry**

**Rating:** 7
**Confidence:** 4
**Correctness:** Overall, I did not find any obvious i…

**Strengths:**

This work is in a generally good shape. Specifically

-   The analysis provides valuable insights for both AI researchers and chemists on effectively applying LLMs to advance chemistry.
-   Identification of model strengths and weaknesses guides future research directions for improving LLMs for chemistry.
-   The benchmark and findings will catalyze more research on LLMs in chemistry.

**Additional Feedback:**

N/A

**Clarity:**

Some parts could be simplified or explained better, but overall it reads well.There are a few areas where could be clarified further:

-   For text-based molecule generation and captioning tasks, the differences (i.e., prompts/performance) from the MolT5 baseline model could be highlighted more.

**Documentation:**

This is a benchmark study and it is sufficient to support reproducibility.

**Ethics:**

The data sources appear ethically sound. The authors could confirm no personally identifiable or sensitive information is included.

**Limitations:**

1. No ablation study on the impact of different prompt components.
2. Comparison to baselines trained on full datasets is not fully fair.

**Opportunities For Improvement:**

1. It would be interesting to see how performance evolves across different versions of GPT-4 on these chemistry tasks. Evaluating multiple GPT-4 versions could provide insights into progress in model capabilities.
2. The paper could benefit from more analysis into why LLMs selectively outperform baselines on certain property prediction tasks. Providing the actual prompts used may clarify how property labels are leaked. More interpretation of these unexpected results is needed.
3. Including additional name prediction baselines could provide more context on the relative performance of LLMs. The claims around lines 215-217 regarding struggles with SMILES representations could also be supported with more concrete evidence.
4. Retrieval and prompt engineering are key modules in the ICL. The paper would be strengthened by a more exhaustive exploration and ablation study on these factors. This is noted as useful future work.
5. Discuss potential negative impacts of generating harmful chemicals.

**Relation To Prior Work:**

Overall, the paper provides a reasonable overview of how this contribution differs from related works through statements in the introduction and related work sections.

**Summary And Contributions:**

This paper presents a comprehensive benchmark for evaluating Large Language Models (LLMs) such as GPT on practical chemistry tasks. The key contributions are:

-   Identification of three core chemistry capabilities to evaluate in LLMs: understanding, reasoning, explaining; Evaluation of major LLMs - GPT-4, GPT-3.5, Davinci-003, both zero-shot and in-context learning
-   Formulation of eight representative chemistry tasks covering these capabilities: Name prediction, property prediction, yield prediction, etc.
-   Systematic experimental framework accounting for: Impact of prompt design and in-context learning; Model randomness through 5 repetitions; Grid search on validation set to select optimal prompts
-   Broad analysis of model capabilities and limitations on chemistry problems; as well as insights on how different factors affect LLM performance on chemistry tasks

Overall, the work provides a comprehensive benchmark and analysis of LLMs on practical chemistry problems. It highlights strengths and weaknesses of current models, while also showing promising potential. The benchmark and findings will inform future research on applying LLMs to advance chemistry.

---

> ### Author Response · Authors · 2023-08-22
> **Response to Reviewer nBsG[1/2]**
>
> We thank the reviewer for their positive feedback and detailed review. Please find our responses to your questions and concerns below, which we also have implemented together within the updated version of the paper.
>
> **Q1:It would be interesting to see how performance evolves across different versions of GPT-4 on these chemistry tasks. Evaluating multiple GPT-4 versions could provide insights into progress in model capabilities.**
>
>
> > A1: Thank you for the interesting suggestion. We acknowledge that a systematic  evaluation across various versions could be conducted by using a consistent set of metrics and test datasets for all versions. However, it cannot be done since different versions of GPT-4 have been trained with varying datasets, configurations, and objectives which Openai has not yet made this technical detail publicly. Thus, direct comparisons will be  challenging.
>
>  **Q2: The paper could benefit from more analysis into why LLMs selectively outperform baselines on certain property prediction tasks. Providing the actual prompts used may clarify how property labels are leaked. More interpretation of these unexpected results is needed.**
>
> > A2: We agree with the reviewer that this is one of the most interesting findings of the work and have elaborated on it as shown below.  For the HIV and ClinTox datasets, in addition to the SMILES strings and their corresponding labels, we also incorporate information that is closely tied to the final prediction. Specifically, for the HIV dataset, we consider the activity test results. Molecules are very likely to inhibit HIV replication if the activity test is labeled as "confirmed active" or "confirmed moderately active." For the ClinTox dataset, whether a molecule is FDA-approved serves as an indicator of its clinical toxicity. There's a high likelihood that a molecule is clinically toxic if it isn't FDA-approved. We conducted experiments where we removed these contextual pieces of information from the in-context learning prompts. Our results, presented in Table 1 and Table 2, show that removing related context indeed leads to a decrease in prediction accuracy.
>
> > Table 1: Impact to F1 score of removing label context information from the in-context learning prompts.
> > |         F1(↑)                | HIV               | ClinTox           |
> > | ---------------------------- |:----------------- | ----------------- |
> > | GPT-4(zero-shot)             | $0.977\pm(0.013)$  | $0.489\pm(0.018)$|
> > | GPT-4(unlabelled, zero-shot) | $0.554\pm(0.017)$ | $0.438\pm(0.045)$ |
> > | GPT-4(few-shot)              | $0.797\pm(0.021)$ | $0.563\pm(0.008)$ |
> > | GPT-4(unlabelled, few-shot)  | $0.493\pm(0.030)$ | $0.478\pm(0.035)$ |
>
> > Table 2: Impact to accuracy of removing label context information from the in-context learning prompts.
> > |              Accuracy(↑)     | HIV               | ClinTox           |
> > | ---------------------------- |:----------------- | ----------------- |
> > | GPT-4(zero-shot)             | $0.986\pm(0.070)$  | $0.736\pm(0.027)$|
> > | GPT-4(unlabelled, zero-shot) | $0.628\pm(0.016)$ | $0.602\pm(0.039)$ |
> > | GPT-4(few-shot)              | $0.836\pm(0.020)$ | $0.856\pm(0.014)$ |
> > | GPT-4(unlabelled, few-shot)  | $0.541\pm(0.032)$ | $0.630\pm(0.014)$ |
>
> > We have updated the result to the new session - "Label interpretation" in Appendix B (Molecule Property Prediction).
>
>
> **Q3: Including additional name prediction baselines could provide more context on the relative performance of LLMs. The claims around lines 215-217 regarding struggles with SMILES representations could also be supported with more concrete evidence.**
>
> > A3: In response to the reviewers suggestion, we added the STOUT (Smiles TO iUpac Translator)[1] as the name prediction baseline. The results are shown below and demonstrate that the performance of LLMs is still inferior to the baselines.
>
> >|  Method  | smiles2iupac | iupac2smiles | smiles2formula | iupac2formula |
> >|:--------:|:------------:|:------------:|:--------------:|:-------------:|
> >| STOUT[1] |     0.55     |      0.7     |        -       |       -       |
>
>
> > We will add more concrete examples in the next version of the paper to show the LLMs struggle with SMILES representations.
> >
> > [1] Rajan, Kohulan, et al. STOUT: SMILES to IUPAC names using neural machine translation.

---

> > ### Author Response · Authors · 2023-08-22
> > **Response to Reviewer nBsG[2/2]**
> >
> > **Q4:Retrieval and prompt engineering are key modules in the ICL. The paper would be strengthened by a more exhaustive exploration and ablation study on these factors. This is noted as useful future work.**
> >
> > > A4: Thanks for this insightful advice. Designing advanced prompting techniques for **each task** to evaluate the performance of **each LLM** would be very challenging and we regard it as our long-term goal. Since advanced prompting highly depends on the specific LLMs to use on specific tasks, many previous famous LLM benchmark work such as BIG-Bench [1] and Chain-of-thought hub [2] also used prompts based on the best of experts' knowledge, and leave the design of advanced prompting for future study.  Moreover, some advanced prompting ways may be helpful for one LLM but hurt the performance of others. Thus, this would cause unfair evaluation in the benchmark. Due to these challenges, our prompting design considers the key factors: fair and general for most LLMs, human easy-written and understanding, producing stable and best possible results.
> >
> > > Based on this, in our previous experiments, we have tried different kinds of prompts on our validation set. Our prompting is a **trade-off between "general for most LLMs" and "producing stable and best possible results"**. According to the categorization of LLM prompts  in  [3], our prompting is in the most detailed level which consists of role-play description, task description, guidance of the few-shot examples, etc. We thus believe that  our prompting can achieve best possible results, and also is fair and general for most LLMs.
> >
> > > Overall, we appreciate for this advice and we will continually update our prompting to build a better and fair LLM benchmark for chemistry.
> >
> > > [1] A Srivastava, et al. Beyond the Imitation Game: Quantifying and extrapolating the capabilities of language models.
> > [2] Yao Fu, et al. Chain-of-Thought Hub: A Continuous Effort to Measure Large Language Models' Reasoning Performance.
> > [3] Shubhra Kanti Karmaker Santu et al. TELeR: A General Taxonomy of LLM Prompts for Benchmarking Complex Tasks
> >
> > **Q5:Discuss potential negative impacts of generating harmful chemicals.**
> > > A5: Thanks for your important suggestion. We have added this discussion as Section 5.5 in the paper. We are aware of active discussion about this concern in a number of venues. We added  session-5.5 under *Discussion* to highlight the capability of GPT models to generate chemically valid molecules, which can make positive contribution such as new medicines and sustainable materials. At the same time, any technology as powerful as LLMs has the potential risks of misuse, where such AI technologies can be repurposed to produce harmful, toxic, or prohibited substances. Therefore, we believe that rigorous oversight, robust safeguards, and ethical frameworks to prevent detrimental applications of these tools is needed for the AI for science community. These obviously need to be implemented by the developers of LLMs.
> >
> > **Q6: For text-based molecule generation and captioning tasks, the differences (i.e., prompts/performance) from the MolT5 baseline model could be highlighted more.**
> > > A6: Thanks for the suggestion. We highlighted the differences in Appendix G (Text-based Molecule Design) and Appendix H (Molecule Captioning).

---

> > > ### Comment · Reviewer_nBsG · 2023-08-25
> > > **Thanks for the update**
> > >
> > > Thanks for the update, I'll remain my positive score. And looking forward to how these LLMs could facilitate science as OpenAI release the fine-tuning APIs, so we now have more tools other than ICL.

---

> > > > ### Author Response · Authors · 2023-08-27
> > > > **Thank you**
> > > >
> > > > Thank you for your thoughtful review and feedback, we truly appreciate your time and expertise.

---

### Author Response · Authors · 2023-08-22
**Global Response: Summary of Paper Changes**

Dear reviewers and AC,

We would like to thank the reviewers for   their insights that offer us great advice towards writing a more comprehensive and competitive paper! As reviewers highlighted, our work studies the application of LLMs for a range of chemistry applications, elucidating their strengths and weaknesses in a range of tasks. We are encouraged that the reviewers agree that "the analysis provides valuable insights", "covers a broad range of tasks" on "a very interesting topic", is "well presented" and provides a "high-quality guide for researchers". We have also made revision sincerely to address all the reviewers’ concerns, and provide detailed answers to each of your comments below. Given the enhancements based on the feedback, we believe our work now resonates even more profoundly and meets a heightened standard, poised to inspire a wider audience.


Thank you very much.

with best Regards,

Authors

---

### Decision · Program_Chairs · 2023-09-22

**Decision:**

Accept (Poster)

**Comment:**

This paper performs an in-depth analysis of the capabilities of existing LLMs (GPT-4 and GPT-3.5 included) in the chemistry domain. The authors proposed three types of chemistry understanding tasks and set up 8 data sets to evaluate the capabilities of LLMs on these tasks. The codes and data sets are open-sourced. All reviewers agree this paper is a good contribution of LLMs for chemistry understanding.